# Calcite carbonate sinks low-density plastic debris in open oceans

Xiang-Fei Sun[1,2,4], Yanxu Zhang [3,4], Meng-Yi Xie[1], Lei Mai [1,2] &
Eddy Y. Zeng [1,2] ✉

The vertical settling of plastic debris in oceans is poorly understood. A large share of low-density microplastics (LDMPs) are largely absent from sea surfaces. The present study employs a model that considers the potential of an overlooked microbially induced calcium carbonate precipitation (MICP) process and new motion equations for irregular LDMPs. Here we show that the motion of LDMPs in the present model, exhibiting a damped oscillation pattern, is quite different from that in biofouling models. Furthermore, LDMPs in the size range of 10–200 μm are most likely to gain sufficient density at the biofouling/MICP stage to independently sink to the ocean floor with relatively small drag coefficients, potentially explaining the selective enrichment of LDMPs in the oceanic sediment. The size and shape exhibit strong non-linear effects on the settling patterns of LDMPs. Overall, the present study highlights the importance of calcite-mediated sinking of LDMPs in open oceans.

Marine plastic pollution has been a global concern, but plastic fate in deep seas has remained mysterious[1]. Low-density plastics, such as polyethylene (PE) and polypropylene (PP), account for over 50% of the total plastic waste, which can float on the sea surface, drift through the ocean currents, and spread into the global ocean[2]. Plastic debris conform to a fractal process through aging and fragmentation upon photolyzation and abrasion, breaking into smaller pieces[3]. However, field measurements confirmed that the estimated amount of low-density plastic debris discharged into the global ocean was much greater than that observed on the oceanic surface, especially for plastic debris with size below 5 mm, also known as low-density microplastics (LDMPs)[4,5]. For instance, LDMPs in the range of 0.33–1.00 mm were roughly 40% fewer in numbers than larger LDMPs (1.01–4.75 mm)[6]. "Missing plastics" and "lost plastics" have led to broad investigations into the potential inventories or sinks of LDMPs in the ocean[7].

Although field investigations regarding deep-sea LDMPs are still challenging, scientists have managed to obtain samples of deep ocean water and sediment from multiple locations since 2013[8–10]. Considerable amounts of LDMPs were detected from the twilight zone to the abyssal seafloor[11,12]. In seawater, LDMPs were detected at the 1000m depth, peaking at 200 – 400 m at an off-shore site close to California, USA[13]. Li et al.[14] collected LDMPs at a spate of vertical depths up to 2000 m at six locations in the Pacific Ocean and the Indian Ocean. Nakajima et al.[15] also reported the widespread occurrence of benthic plastic debris in the deep-sea basin of the Northwest Pacific (3500–6500 m), up to 70% of which are LDMPs. In oceanic sediment, abundant LDMPs were found at the Southwest Indian Ocean (900-1000 m)[8], the Northeast Atlantic Ocean (1400–2200 m)[8], the Arctic Ocean (2783–5570 m)[9], and the western Pacific Ocean (5108–10,908 m) at the Mariana Trench[10]. A tentative "whole ocean" mass balance theory suggested that a large share of LDMPs have settled on the seafloor[7], but rational explanations on LDMPs vertical settling remain challenging. Laboratory tests were deemed impossible to simulate LDMP settling in the deep ocean[16], and modeling had become a viable alternative. Kooi et al.[17] built the first idealized 1D, depth-dependent LDMP vertical settling model, considering biofouling as the density shifter[18]. Simulated LDMPs exhibited a repeated oscillation pattern near the upper water column[17]. Although the outcome has never been observed[19], the biofouling model could explain the low concentration of LDMPs in the ocean surface[4], as well as the accumulation of LDMPs within 200 m beneath the ocean surface[12].

[1]School of Environment and Energy, South China University of Technology, Guangzhou, China. [2]Southern Marine Science and Engineering Guangdong Laboratory (Zhuhai), Zhuhai, China. [3]School of Atmospheric Sciences, Nanjing University, Nanjing, China. [4]These authors contributed equally: Xiang-Fei Sun, Yanxu Zhang. ✉e-mail: eddyzeng@scut.edu.cn

However, the biofouling model encounters an enormous challenge to explain the enrichment of LDMPs in the oceanic sediment. Modeled LDMPs were impossible to approach the ocean floor due to the continuing loss of biofilm[20]. Fischer et al.[19] evaluated LDMP settling under another biofilm dynamic, assuming frustule attachment on LDMPs after settling beneath the twilight zone. Ingestion by marine organisms (fecal pellets) merely sinks a negligible fraction (0.13–0.19%) of LDMPs[21]. Aggregation with biogenic particles (marine snow) could sequestrate another 0.06–8.8% of LDMPs[22]. Lobelle et al.[23] and Fischer et al.[19] incorporated oceanographic processes with a settling model, including large-scale 3D advection, small-scale vertical turbulence, dynamic grazing, and wind-induced mixing etc. The updated model allowed LDMPs to sink below the euphotic zone and mixed layer; however, only 15 of 10,000 LDMPs reached the ocean floor (>5000 m) under favorable conditions[19].

Microbially induced calcium carbonate precipitation (MICP) is commonly observed under calcium-rich and high pH environments, such as ocean surface and soil matrix[24,25]. The present study considers MICP accompanied by biofouling (autotrophic path)[26]. Most algae have negative surface charges[27], and continue attracting oversaturated $Ca^{2+}$ in the upper seawater column[27,28]. Some biological processes, such as photosynthesis and hydrolysis of urea, can increase pH levels in surrounding water[25], triggering $CaCO_3$ to precipitate near the biofilm and become part of the biofilm exoskeleton[24,29]. Different types of $CaCO_3$, including calcite and aragonite[30], can be produced by MICP. However, aragonite formation requires temperatures over 30 °C[31]. Thus, calcite is considered the primary precipitate on LDMPs.

With a density over 2.63 g cm$^{-3}$, calcite could perform as an effective ballast to sink LDMPs to ocean sediments.

In the present study, a new one-dimensional hydrodynamic model is designed to evaluate the impact of MICP on LDMP settling. The MICP is controlled by the algal photosynthetic intensity. The target LDMPs have a density of 0.85–1.00 g cm$^{-3}$ with an equivalent size range between 1 μm and 5 mm (see "Methods"), following an exponential distribution[32]. Sphere, fiber (57–13,000 μm in length), and film (0.1–100 μm in thickness) are evenly distributed. New hydrodynamic equations are introduced to improve the model's performance on irregular LDMPs. The settling dynamics of LDMPs with and without MICP are compared, and the impact of size and shape on settling behaviors of LDMPs are discussed. The observed enrichment of LDMPs in the ocean subsurface and sediment is also discussed by simulating the vertical settling of a group of randomly generated LDMPs using the Monte Carlo approach (see "Methods").

## Results and Discussion

### Impacts of calcite precipitation on the settling of LDMPs

The essence of LDMP settling is to increase its density by binding with high-density substances, and heavy and steady fouling is always beneficial. Calcite (2.63 g cm$^{-3}$) is an efficient ballast in increasing and maintaining the density of LDMPs compared to biofilm (1.15–1.18 g cm$^{-3}$)[18], algal frustules (1.80 g cm$^{-3}$)[33], marine snow (1.02–1.03 g cm$^{-3}$)[20], and fecal pellets (1.02–1.06 g cm$^{-3}$)[34]. A 10 μm sphere PE with an initial density of 0.92 g cm$^{-3}$ only needs 0.35 μm thick calcite precipitates to approach a density of over 1.1 g cm$^{-3}$, which has already exceeded the highest known seawater density of 1.09 g cm$^{-3}$ [35]. Moreover, calcite precipitates are difficult to come off plastic surfaces[25]. Laboratory studies revealed that the binding between calcite and LDMPs is so strong that calcite can be used as a coat to strengthen plastics in cementitious materials[36].

To fully evaluate the settling patterns of LDMPs, we use the Monte Carlo method to generate 500 random LDMPs, calculate their vertical trajectories with and without MICP using the present model, and record their depths in a time series of 200 days (Fig. 1). The settling patterns of LDMPs in the deep ocean can be substantially altered by MICP, as compared to the results by Kooi's model, which only considered biofouling process (Fig. 1A). Shortly after being released to simulated seawater, algae begin to collide and attach on the LDMP surface. Depending on the algal concentration and kernel encounter rate[2,37], biofilm forms within minutes to hours[38]. During the process, MICP delivers calcite on the plastic surface. With growing ballast (biofilm and calcite), the density of LDMPs surpasses the density of surrounding seawater, triggering initial settling of LDMPs to the water column. Due to light intensity attenuation with increasing depth, biofilm decays through dying, shrinking, and shedding[39]. Although MICP also fades due to reduced pH caused by weakened algal photosynthesis[24], calcite precipitates are more difficult to detach from the plastic surface than biofilms. Thus, LDMPs can retain most of the gained density and continue to settle until the negative buoyancy is balanced with the vertical drag. Depending on the settling conditions, most LDMPs could sink to the seafloor within tens to hundreds of days (Fig. 1B). Microbially induced calcium carbonate precipitation may serve as one of the critical factors in depleting LDMP concentrations at the ocean subsurface and enhancing LDMP accumulation at the oceanic sediment[12].

The present model also considers the impact of calcite dissolution during LDMPs settling, especially after penetrating the calcite saturation depth (CSD)[24]. Seawater saturation state and oxygen consumption are the driving forces behind calcite precipitation or dissolution[40]. The level of calcite saturation decreases in cold deep water. Because the solubility of calcite increases with increasing seawater depth/pressure, the concentration of dissolved inorganic carbon is elevated relative to total alkalinity levels in the deep ocean[41]. Oxygen consumption due to

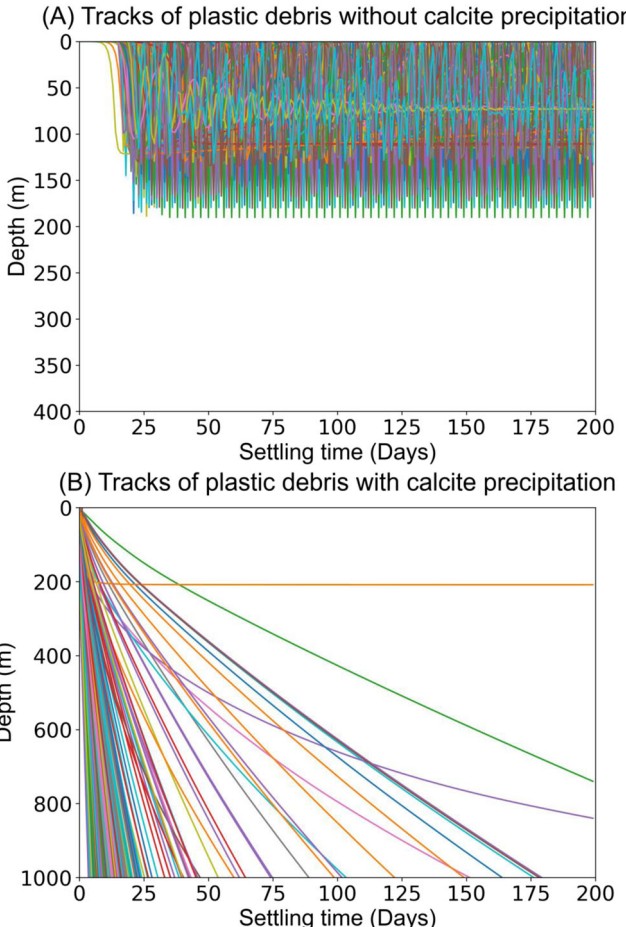

**Fig. 1 | Simulated trajectories of typical low-density microplastics (LDMPs) in time series during vertical settling in the tropical Pacific Ocean. A** Without microbially induced calcium carbonate precipitation (MICP). **B** With MICP.

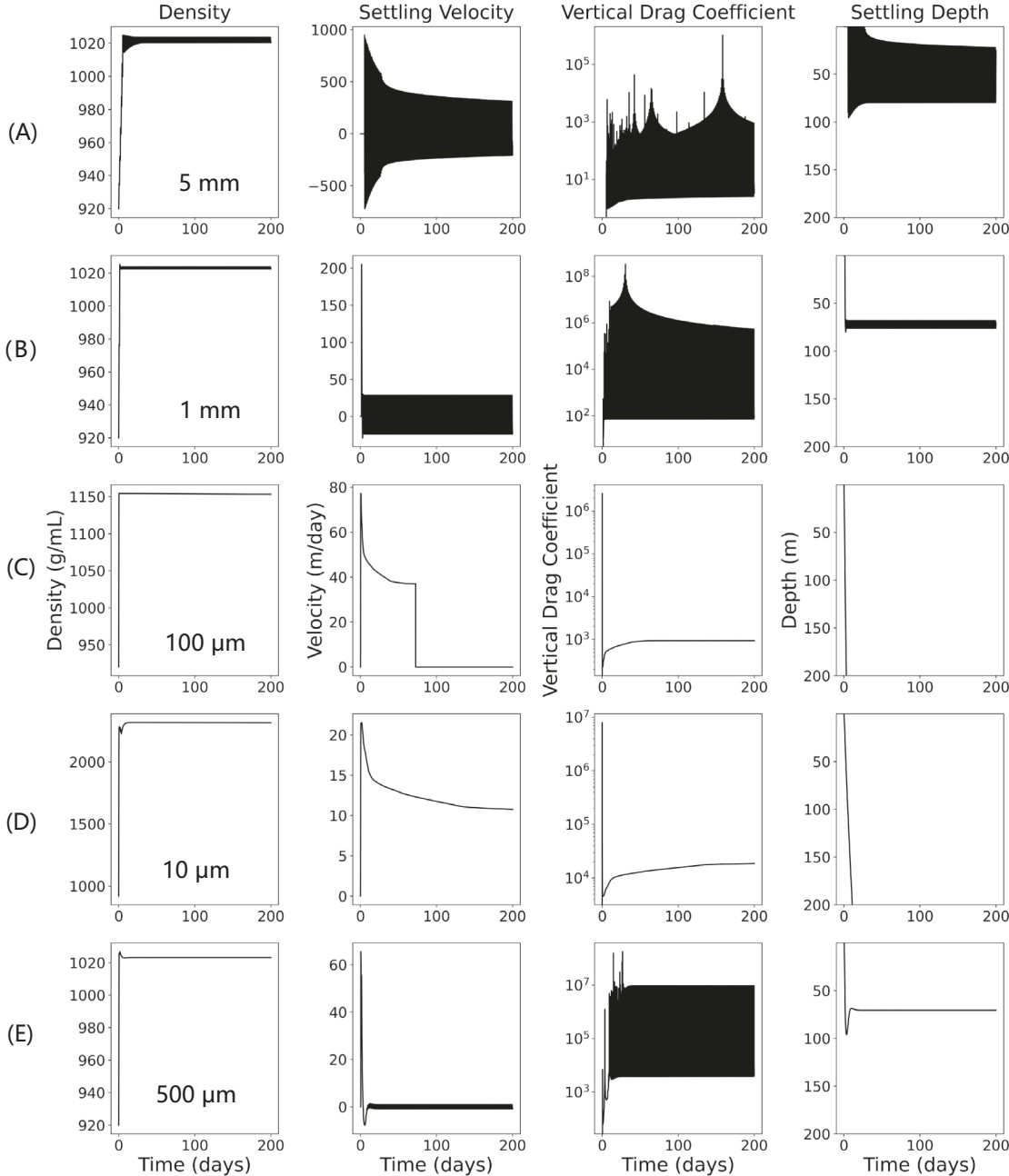

**Fig. 2 | Simulated density, settling velocity, vertical drag coefficient, and settling depth of typical low-density microplastics (LDMPs) in time series with microbially induced calcium carbonate precipitation under the tropical Pacific** **Ocean conditions.** For comparison, the simulated LDMPs are in perfect sphere shape with shape factor equal to 1. The sizes of LDMP: **A** 5 mm; **B** 1 mm; **C** 100 μm; **D** 10 μm; and **E** 500 μm.

biological respiration generates carbon dioxide and organic acids, enhancing calcite dissolution[40]. Yet, our model simulation indicates that most LDMPs only suffer slight calcite loss and settling velocity deceleration, even after passing the CSD. The impact of calcite dissolution is relatively small for three reasons. First, oxygen consumption is largely inhibited during the MICP process because of the high pH environment near the LDMP surface created by photosynthesis[25]. Second, the settling velocity of LDMPs is relatively high. Previous studies reported little $CaCO_3$ dissolution was observed on particles at a canonical sinking rate of 100 m day⁻¹, so the sinking flux at 4000 m is almost identical to the surface flux[40]. Based on our model simulation, most LDMPs can reach the sea floor within less than one year, leading to insufficient dissolution. Third, the CSD varies regionally from 1000 m to 4600 m[24]. Calcite dissolution is limited when the seafloor is

above the CSD. For instance, the CSD in most regions of the Atlantic Ocean is below 3500 m[24]. In comparison, the average depth of the Atlantic Ocean is 3646 m.

## Two settling patterns of LDMPs
Depending on how much calcite precipitates remain and how the seawater density varies, LDMPs can either undergo damped oscillation at the epipelagic zone or settle into the oceanic sediment (Fig. 2). Using the tropical Pacific Ocean profiles as an example, we conducted a series of sensitive analyzes to study the two settling patterns of LDMPs by tracking the trajectory, settling velocity, vertical drag coefficient, and density of a series of typical spherical LDMPs.

For relatively large LDMPs with a shape factor close to the sphere (the smallest specific surface area under the same volume), the

damped oscillation pattern is frequently observed in the model simulation results (Fig. 2A, B). For large LDMPs with relatively small specific surface areas, the vertical drag coefficients are small. A slightly higher density of large fouled LDMPs than that of surrounding seawater would trigger the settling process (e.g., > 500 μm spheres), with a settling velocity over 200 m day$^{-1}$. A slight loss of biofilm during settling would lead to density reduction, creating positive buoyancy. With a small vertical drag coefficient, LDMPs would rise quickly toward the sea surface, where biofilm and calcite can grow and accumulate again. Unlike the biofilm model, LDMP cannot approach its previous depth due to the existing calcite participates, resulting in a damped oscillation. During this time, calcite precipitates would slowly accumulate on each cycle. It is speculated that LDMPs would reach a suspension state when nearly no algae exist on the LDMP surface or are substituted by other microorganisms.

In contrast, the vertical drag coefficient of small (< 500 μm) and non-spherical LDMPs (e.g., film, fragments, and fiber) is quite large, which could slow down the LDMP motion (Fig. 2C, D). Established biofilms and calcite precipitates can be developed on the LDMP surface, leading to a much higher density of fouled LDMP than seawater density. In some cases, the density can grow up to 2.40 g cm$^{-3}$ (Fig. 2D) before fouled LDMPs could leave the epipelagic zone. For fouled LDMPs with established biofilms and calcite precipitates, the vertical settling is straightforward, because the extra ballast of calcite could assist LDMPs to overcome seawater density gradient and resist calcite dissolution (Fig. S1).

The simulation results also indicate a transitional state between the two settling patterns (Fig. 2E). With proper vertical drag coefficient, the LDMP settling would neither follow an oscillation pattern nor sink quickly to the ocean floor. These particles would accumulate just enough biofilm and calcite to gain negative buoyancy, but they cannot maintain negative buoyancy upon increased seawater density and biofilm loss. Furthermore, the LDMP density with remaining calcite precipitates and biofilms balances with the seawater density, allowing LDMPs to enter the suspension state directly without other oceanographic processes involved.

## Impacts of MICP on sinking patterns of LDMPs

The settling of LDMPs is controlled primarily by negative buoyance created by biofilm and calcite, as well as the vertical drag coefficient affected by the physical properties of LDMPs. We evaluate the initial density, shape, and size of LDMPs and their impact on settling patterns. The shape categories of LDMPs (Fig. 3A–I) are evenly assigned as sphere, film, and fiber. Simulation results indicate that the settling of LDMPs exhibits essentially no relevance to the initial density of LDMPs, which is determined mainly by MICP and biofilm growth (Fig. 3A–C).

The shape of LDMPs plays an important role in determining the specific surface area of LDMPs (Fig. 3D–F). Fibers and films have larger specific surface areas than spheres, leading to higher vertical drag coefficients (Fig. 4A). Under the same volume, spheres are more sensitive to negative buoyancy with larger settling velocities than films and fibers. Due to the high vertical drag coefficient, films and fibers have relatively enough time for biofouling and MICP and therefore higher densities than spheres. Spheres are more concentrated at 100–200 m, where fibers and films distribute in a much deeper and broader region of 100–400 m (Fig. 3D–F). The wider distribution of fibers and films compared to spheres can be explained by the uncertainty of the shape factor due to the extra degree of freedom of motion. Unlike spheres, fibers and films can twist and flip naturally during settling, so the projection area towards the settling direction could vary accordingly, which creates deviations in the vertical drag coefficient[42]. Our simulation also exhibits that fibers and films require a much longer time to reach a steady state than spheres. Sphere-shaped LDMPs are most likely to accumulate on the seafloor. A close

examination of simulated LDMPs sized below 1 mm indicates that fibers and films need much longer time than spheres to reach the seafloor, especially for those with equivalent diameters smaller than 100 μm. This trend has been observed in several previous studies, and the present model may provide a viable explanation for the high abundance of fibers and films in the water column[43].

The size of LDMPs is another crucial factor governing the specific surface area (Fig. 3G–I). A larger size with the same shape category means a smaller specific surface area, which corresponds to a lower vertical drag coefficient and density difference and higher settling velocity. For spheres (shape factor close to 1), LDMPs at a size range below 500 μm could stack enough calcite ballast to settle into the oceanic sediment without aggregation with fecal pellets and marine snow[44] (Fig. 4B). The maximum sizes for these sinking fibers and films can be one order of magnitude larger than those for sinking spheres at equivalent diameters (equivalent in volume to sphere diameter). When LDMPs sink below the minimum light intensity depth, their density can approach at least 1.04 g cm$^{-3}$ for a 500 μm sphere (up to 2.60 g cm$^{-3}$ for nanoplastics; Fig. 4C). It is noted that the upper size limit for LDMPs to independently sink to the seafloor depends on the specific seawater conditions. The current upper size limit of 500 μm is only for spherical LDMPs settling under the tropical Pacific Ocean profile, where seawater has a maximum density of 1.035 g cm$^{-3}$.

Sphere-shaped LDMPs in a size range of 100–500 μm can directly settle to the ocean sediment, and their settling velocities (100–300 m day$^{-1}$) are much higher than marine snow (68 m day$^{-1}$)[45] and fecal pellet (25–67 m day$^{-1}$)[46]. For sphere-shaped LDMPs in the size range of 10–100 μm, the settling velocity is reduced to 10–20 m day$^{-1}$. Increased vertical drag coefficient for LDMPs in the size range below 10 μm would further decrease the settling velocity to less than 1 m day$^{-1}$ (Fig. 4D). The high abundance of LDMPs in the simulated water column with the size range below 10 μm is inconsistent with field measurements. Moreover, LDMPs in the size range of 1–5 mm cannot settle to the seafloor, which is consistent with the observed size distribution of LDMPs at the ocean sediment.

In previous studies, aggregation has been demonstrated as one of the important processes assisting the settling of ocean particles[37], which may also boost the settling of LDMPs with the size ranges of <10 μm and 1–5 mm[26,47]. The simulation results indicate that such aggregation could sink a significant amount of LDMPs in the size range below 10 μm, aggregating them with portions of LDMPs in the size range of 1–5 mm. On one hand, oscillating large-sized LDMPs have more chances to collide with other LDMPs. On the other hand, small-sized LDMPs are much more abundant according to the exponential size distribution of LDMPs. With high-density fouling, these small-sized LDMPs are the perfect ballast to aggregate with large-sized LDMPs. Aggregation of LDMPs with marine snow and fecal pellets also occurs, and the aggregates could settle to the seafloor under a small vertical drag coefficient.

## Two sinking patterns of LDMPs

Figure 5 summarizes the field measurements of LDMPs in the seawater column (13 sampling sites) and oceanic sediment (6 sampling sites) in the Indian Ocean, Pacific Ocean, Atlantic Ocean, and Arctic Ocean[13,14,48,49]. High concentrations of LDMPs were found in the subsurface water column and oceanic sediment. With new motion equations, MICP, and designated ocean data from the National Oceanic and Atmospheric Administration (NOAA)[50], the present model has successfully reproduced the distribution patterns of LDMPs, explaining the settling dynamics behind the field observations, especially the enrichment of LDMPs in the size range of 10–200 μm in the oceanic sediment.

Under consideration of MICP, two LDMP accumulation zones along the vertical direction are formed. The first accumulation zone occurs at the upper water column near the sea surface (60–400 m),

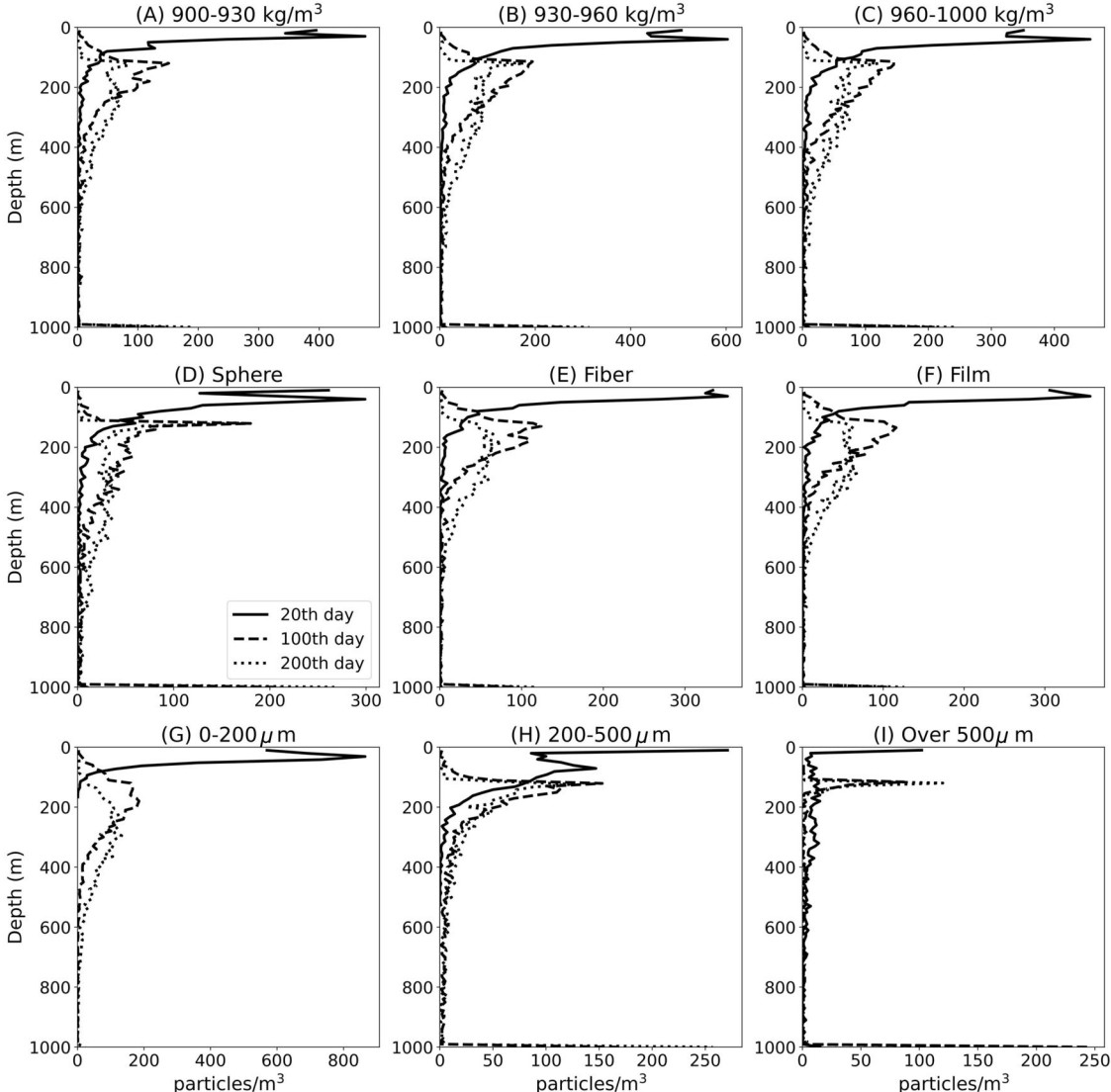

**Fig. 3 | Impacts of initial density, shape, and size on settling of low-density microplastics (LDMPs) in time series.** Vertical distributions of LDMPs at the 20th (solid), 100th (dash), and 200th day (dot) under different initial densities (**A**, **B**, and **C**), shapes (**D**, **E**, and **F**), and sizes (**G**, **H**, and **I**).

coinciding with assumed static seawater conditions. Fischer et al.[19] and Lobelle et al.[23] demonstrated that incorporating oceanographic processes such as large-scale 3D advection, small-scale vertical turbulence, dynamic grazing, and wind-induced mixing could spread LDMPs to much deeper waters (up to 5,500 m), but the maximum concentration of LDMPs remains in the upper water column[51]. It seems that the oceanographic processes can assist in creating dynamic and complex scenarios for LDMPs.

The subsurface accumulation zone commonly appears in subtropical oceans[13,14], where the light intensity and chlorophyll concentration are consistent throughout the year (Fig. 5A–E), creating relatively stable conditions for biofouling and MICP[52,53]. As seawater density increases with increasing depth, it is difficult for large-sized ($>500\,\mu m$) spherical LDMPs to settle as they cannot accumulate enough ballast and are sensitive to density gradients (Fig. 4D). Most of these LDMPs stop settling and concentrate at the subsurface zone[51]. In contrast, the model simulation suggests that LDMPs with a size range below 10 μm can accumulate much heavier ballast (nearly 2.5 g cm⁻³); however, the high vertical drag coefficient traps these LDMPs in the upper water column for a long time due to extremely slow settling velocity[54]. Although aggregation among LDMPs can accelerate the settling process, it is entirely by chance. It is positively correlated with

the concentration of LDMPs in the water column, which further assists in forming the subsurface accumulation zone[26,37,47].

The second accumulation zone (seafloor zone) is at the oceanic sediment. The presence of abundant LDMPs in the oceanic sediment has been confirmed by a number of field investigations[8–10,48,55]. Previous LDMP settling models fail to account for the occurrence of abundant LDMPs in the oceanic sediment. Some of these sites are located far from high-density populations and major river outflows, so LDMPs should have deposited to the sediment from the seawater column. Considering MICP, the present model successfully reproduces the vertical settling of LDMPs, especially for the size range of 10–500 μm under various shapes.

The vertical distribution of LDMPs in the Arctic Ocean is unique compared to those in other oceans, and this unique distribution of LDMPs can be well explained by the present model. The concentration of LDMPs in the Arctic Ocean is higher at both the sea surface and oceanic sediment (seafloor) and lower in the upper water column (Fig. 5A)[1,9,48]. No subsurface accumulation zone is observed, which could be attributed to the dramatic seasonal variations in light intensity and chlorophyll concentration. In summer, light intensity and chlorophyll concentration at the Arctic Ocean reach their maximums, and biofouling and MICP quickly uplift the density of LDMPs (Fig. 5B).

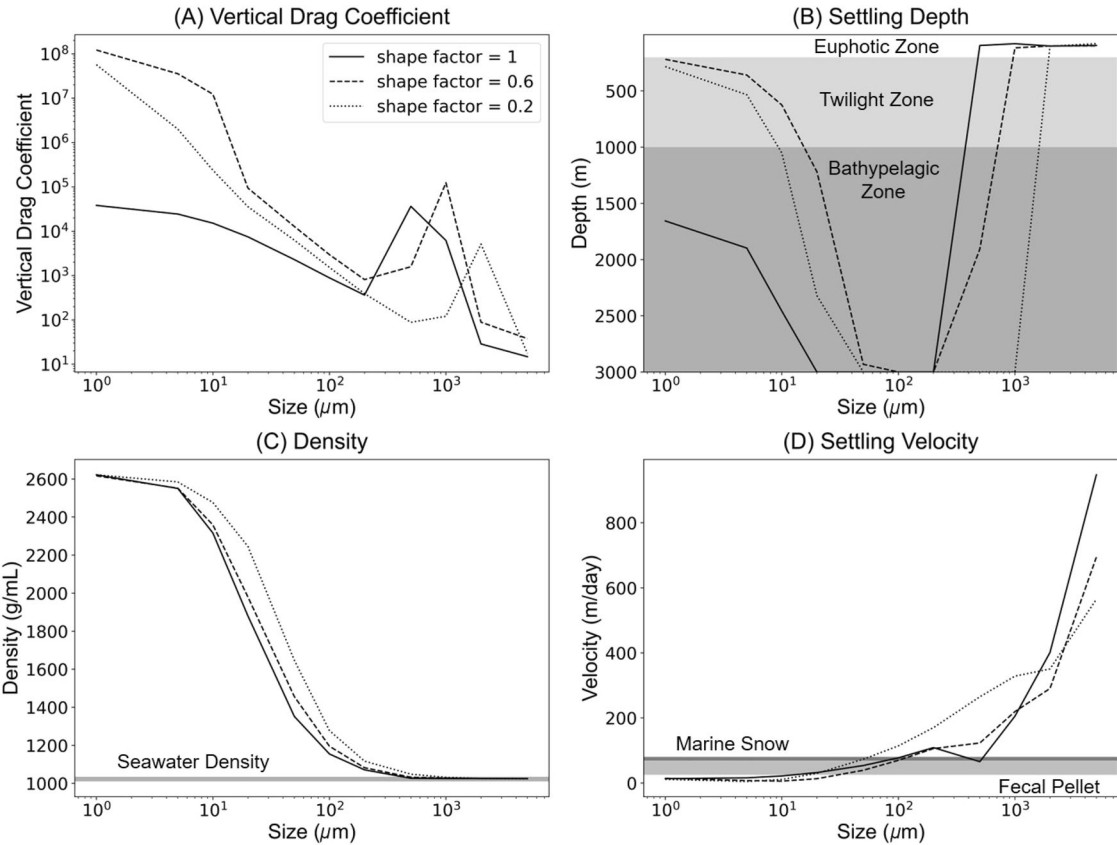

**Fig. 4 | Impacts of shapes and sizes on settling of low-density microplastics (LDMPs) with microbially induced calcium carbonate precipitation after 200 days of settling.** Three shape factors (distinguished in line types) are selected, covering most LDMP shapes. **A** Mean vertical drag coefficients at a given size range of LDMPs. **B**, **C**, and **D** Maximum settling depth, density, and settling velocity, respectively, at each size scale.

Once overcoming the vertical drag coefficient, LDMPs can settle to the seafloor. The seawater density of the Arctic Ocean is almost consistent from the surface to the bottom (Fig. 5C), posing the least hindering effect on the settling of LDMPs[56].

There are also vertical downward currents in the Arctic Ocean, which further promote the settling of LDMPs, especially those with a size below 10 μm. In winter, however, light intensity is almost zero. LDMPs in the subsurface zone would ascend due to the lack of biofouling and MICP, which explains the high abundances of LDMPs on the ocean surface[1] (Fig. 5A, D, E). Seasonal alternation creates a relatively long period of time with no light, which means that the aggregation of LDMPs is insufficient due to inadequate ballast. As a result, the most favorable size range for vertical settling of LDMPs should be 10–500 μm based on the present model, which is consistent with the results from sediment core analyzes[8,9] (Fig. 5F) that the most abundant LDMPs were in the 11–200 μm size class[48]. The aging and fragmentation may also explain the absence of large LDMPs during long-range transport through the global ocean circulation[44,50,57].

**Simulating vertical settling of LDMPs in a time series**

Figure 6 shows the settling distribution of a group of typical LDMPs in a time series with a size range of 1–5000 μm, density range of 0.85–1.00 g cm⁻³, and shape categories of sphere, film, and fiber. The simulation conditions are reported as the tropical Pacific Ocean profiles. The normalized histogram represents the vertical distribution of LDMPs in sphere, film, and fiber. Most LDMPs sink toward deep seas after 30 days of biofouling and MICP. Concentrations of LDMPs at various depths of the upper 1500 m water column decrease dramatically with increasing settling time. The density increment of LDMPs relies on the attached calcite precipitates and biofilms, and the

accumulation of LDMPs in the seafloor requires a considerable amount of time[58,59]. Under the same oceanic conditions, the distribution of LDMPs continuously evolves along the time series.

In the first 20th days, the vast majority of LDMPs remain at the sea surface to a depth of 1500 m, while a few LDMPs have begun to settle into the deep water column (Fig. 6A). At this point, strong size-selective characteristics are observed between the sizes of LDMPs and settling depths (Fig. 3G–I). Fibers and films with equivalent sizes close to those of spheres exhibit similar distribution patterns as spheres. As the settling time passes the 100th day, most LDMPs begin to diffuse into the deep water column, and the subsurface accumulation zone begins to fade away due to lack of continuous supply of LDMPs (Fig. 6B). Spheres occur less commonly in the seafloor than fibers and films. Starting from the 200th day, most LDMPs have already sunk to the seafloor (Fig. 6C). On the 300th day, the distribution of LDMPs shows a gradually dispersed trend, with few LDMPs remaining in the water column (Fig. 6D).

In summary, LDMPs can concentrate in the upper water column beneath the ocean surface and seafloor. LDMPs in the size range of 100–500 μm are more likely to settle independently to the seafloor; settling larger and smaller LDMPs may need to involve aggregation[1,59]. Because efficient aggregation usually requires relatively high concentrations of LDMPs to increase collision probability, LDMPs would stagnate at the epipelagic and twilight zones, boosting the formation of the subsurface accumulation zone. Moreover, the formation of the subsurface accumulation zone seems to rely on a continuous supply of LDMPs from emission sources. Settling of LDMPs on the seafloor creates the second accumulation zone, one of the ultimate sinks for LDMPs. These modeling results are consistent with recent observations[11–14]. Onink et al.[57] also confirmed these patterns through a

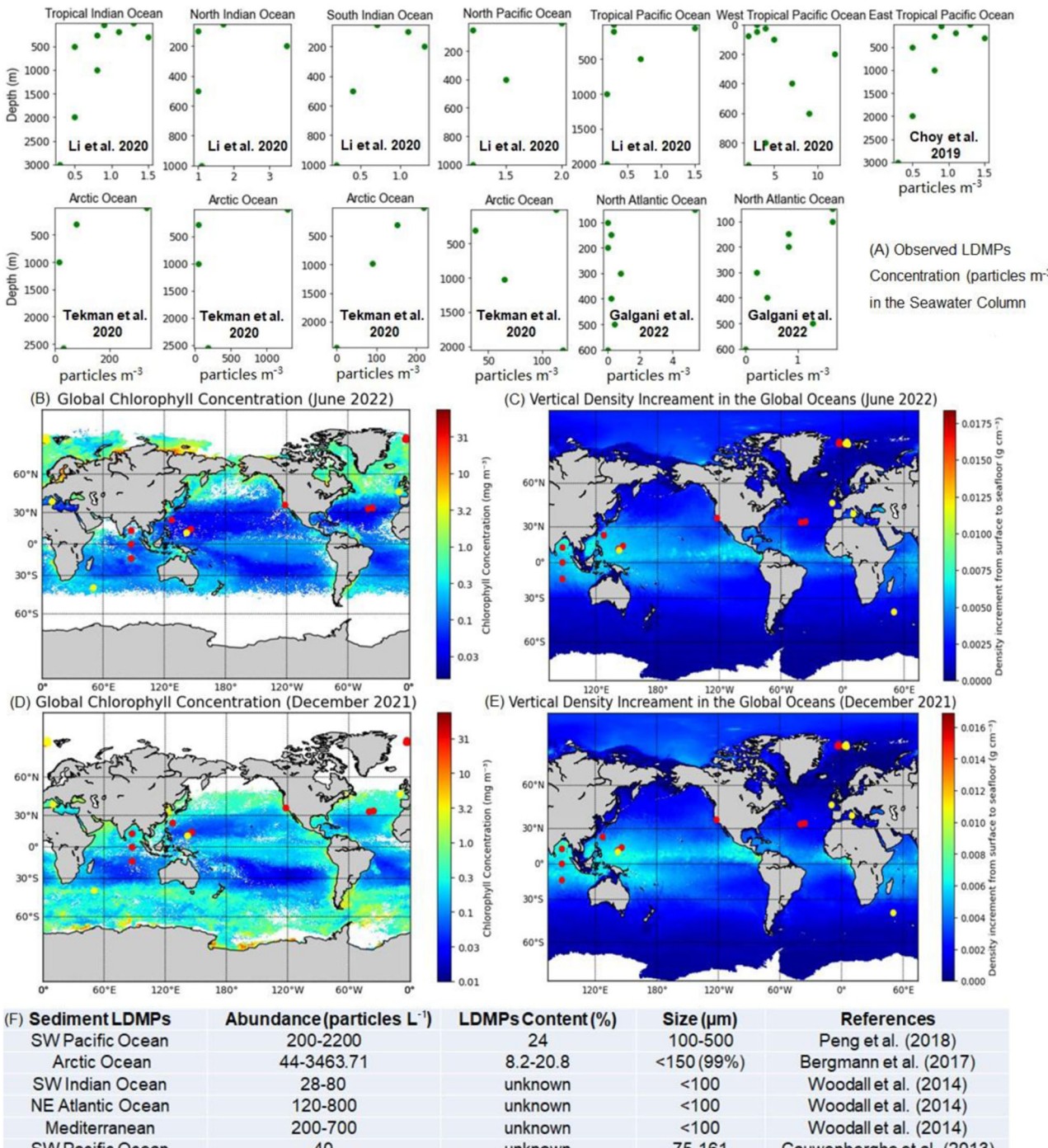

**Fig. 5 | Field measurements of low-density microplastics at the water columns and oceanic sediments in the Pacific Ocean, the Atlantic Ocean, the Indian Ocean, and the Arctic Ocean. A** Previously measured LDMPs concentrations (particles m⁻³) in the water column of different oceans. **B** Global chlorophyll concentration (June 2022). **C** Vertical density increment in the global oceans (June 2022). **D** Global chlorophyll concentration (December 2021). **E** Vertical density increment in the global oceans (December 2021). **F** Previously measured LDMP concentrations (particles L⁻¹) in the sediment of different oceans.

Lagrangian model. The aggregation and settling of small-sized LDMPs into deep waters lead to low near-sea surface concentrations despite their high total abundances.

### Implications for future projections
The present study shows that MICP is vital in the vertical settling of LDMPs in open oceans. It dramatically alters the distribution and motion of LDMPs in the water column. The outcome explains the "missing plastic" or "lost plastic" puzzle in the global ocean, as well as

the abundant and uneven occurrence of LDMPs in the oceanic sediment[4]. The simulation results suggest that the seafloor is likely the ultimate destiny of LDMPs. If LDMPs with calcite precipitates eventually sink to the seafloor, it might impact the ocean benthic ecosystem, global carbon cycle, and ocean calcite cycle. All these potential risks require adequate assessments and systematic studies. We thus call for further strengthening the field measurement of LDMPs in deep-sea water and sediment and searching for LDMPs covered with precipitates under natural conditions. These LDMPs are likely widely

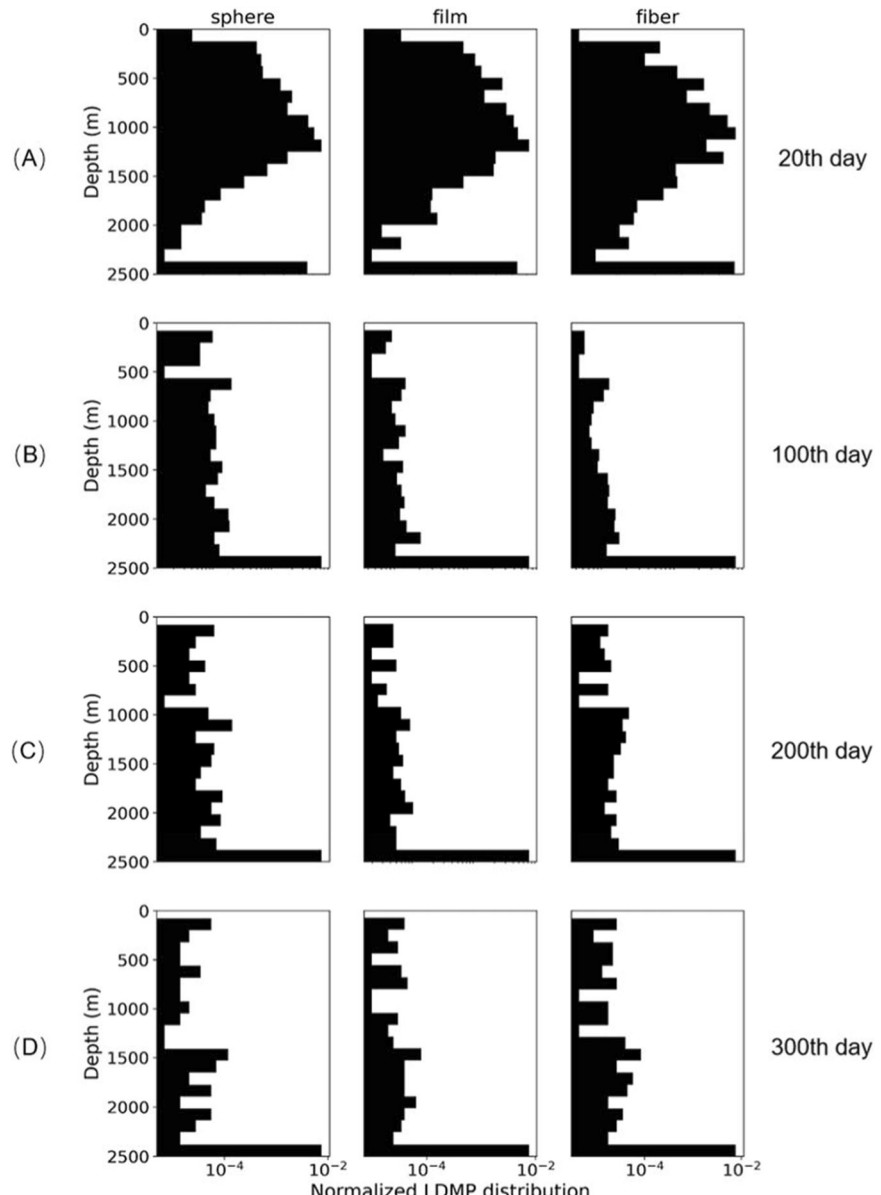

**Fig. 6 | Normalized vertical distribution patterns of mixed sized (10 μm–5 mm) low-density microplastics (LDMPs) with different shapes under selected settling times.** The distribution patterns of LDMPs at different shapes (sphere, film, and fiber) are recorded at the 20th (**A**), 100th (**B**), 200th (**C**), and 300th (**D**) day.

distributed in sediment and deep-sea water and deserve further comprehensive investigations.

## Methods
### Model design principles
The vertical settling of LDMPs is mainly dependent on seawater density, current (vertical and horizontal), nutrition conditions, and biological activity[17,55,58]. The present model calculates the vertical settling speed and trajectory of LDMPs, considering the initial density, shape, and size of LDMPs and designated ocean conditions. The present model is a one-dimensional depth-dependent calcium carbonate ($CaCO_3$) precipitation model using dynamic seawater conditions in a quiescent ocean based on Kooi's model, which is explicitly designed to examine the impact of MICP during plastic settling.

Low-density microplastics under investigation are sized between 1 μm and 5 mm following an exponential distribution, which was adjusted to meet previous observations that the quantity of LDMPs with a size between 100–500 μm is ten times compared to the number of LDMPs generated between 500 μm–5 mm[60]. For fiber-like LDMPs, a

length range between 100 μm and 1.3 mm is selected. The thickness of film-like LDMPs varies between 0.1–100 μm. Each simulated LDMPs is randomly assigned with an initial density from 850 to 1000 kg m$^{-3}$ [61]. A random plastic generator was added to the model, generating LDMPs following the above-mentioned conditions. For simulation purposes, the ratio of plastic debris of different shapes is assumed as fiber: sphere: film = 1:1:1. The model can describe the vertical distribution of a group of random LDMPs using the Monte Carlo method in specific vertical profiles at a given site. The seawater profiles are acquired from the global real-time ocean forecasting system (RTOFS).

The present model includes three complementary sub-models: An updated vertical settling model, a MICP model, and a biofouling model. The vertical settling model based on fluid dynamic principles describes the settling speed and motion of unregulated LDMPs under a wide range of Reynolds numbers, taking into account the temporary density of LDMPs and ocean profiles. The MICP model was designed to estimate calcite precipitates on the plastic surface, considering the activity level of algal photosynthesis and dissolution rate under CSD. The biofouling model simulates biofilm growth on LDMPs, established

by Kooi et al.[17], which includes algae attachment, growth, mortality, and respiration under specific nutrition conditions and light intensity. Key parameters are listed in the supplementary materials (Table S2).

## The vertical settling model
**Motion equations for a wide range of Reynolds numbers.** Previous studies calculated the settling speed or vertical settling velocity using a modified Stokes equation[17], a linearized form of the Navier-Stokes equations for LDMPs with small Reynolds numbers.

$$dz/dt = v_{settling}(z,t) \qquad (1)$$

The Reynolds number may become significantly large for numerous LDMPs with irregular shapes and sizes, and the current Stoke equation may not be accurate for estimation. In 2018, a new one-equation model considering the fluid drag was presented by Dioguardi and Dellino[42], which is suitable for irregularly shaped LDMPs over a wide range of Reynolds numbers:

$$C_d = \frac{24}{Re}\left(\frac{1-\Psi}{Re}+1\right)^2 + \frac{24}{Re}\left(0.1806Re^{0.6459}\right)\Psi^{-(Re^{0.08})} + \frac{0.4251}{1+\frac{6880.95}{Re}\Psi^{5.05}} \qquad (2)$$

$$v_{settling} = \sqrt{\frac{8g(\rho_{pl}-\rho_{sw}^z)r_{tot}^{ESD}}{3C_d\rho_{sw}}} \qquad (3)$$

$$Re = 4\rho_{sw}^z {}^* v_{settling} {}^* r_{tot}^{ESD}/u_{sw} \qquad (4)$$

where $C_d$ is the drag coefficient of LDMPs, $\Psi$ is the shape factor, $Re$ is the Reynolds number, $g$ is the gravitational acceleration (m s$^{-2}$), $\rho_{pl}$ is the density of LDMPs (kg m$^{-3}$), $\rho_{sw}^z$ is the seawater density (kg m$^{-3}$), $r_{tot}^{ESD}$ is the equivalent spherical radius (m), and $v_{settling}$ is the vertical settling velocity (m s$^{-1}$). Equations (2) – (4) show that each one of $C_d$, $v_{settling}$, and $Re$ is affected by the other two factors, which means that an initial estimate of the $Re$ value should be made to start the calculation procedure.

In each cycle, the Reynolds number approaches the actual value. An error estimation function is provided to determine the termination of the calculation:

$$\tau = \frac{Re_n - Re_{n-1}}{Re_{n-1}} \ (\tau < 0.001) \qquad (5)$$

The density of LDMPs $\rho_{pl}$ at the given moment $t$ can be calculated as below:

$$\rho_{tot}(t) = \frac{m_{pl} + m_{bf}(t,z) + m_{Ca}(t)}{V_{pl} + V_{bf}(t,z) + V_{Ca}(t)} \qquad (6)$$

where $\rho_{tot}(t)$ is the density of the LDMP and its attachment (kg m$^{-3}$), $m_{pl}$ is the mass of pure LDMP (kg), $m_{bf}(t,z)$ is the mass of biofilms (algae only, kg), $m_{Ca}(t)$ is the mass of calcite precipitates (kg), $V_{pl}$ is the volume of pure LDMP (m$^3$), $V_{bf}(t,z)$ is the volume of biofilms (algae only, m$^3$), $V_{Ca}(t)$ is the volume of calcite precipitates (m$^3$).

## Estimation of the shape factor
A previous study indicated that the partial hydrodynamic area is critical in settling LDMPs, closely related to the projection area towards the settling direction[62]. A new shape factor is introduced from a recent study considering irregular-shaped particles[62]. The shape factor $\Psi$ is defined as:

$$\Psi = \Phi/\chi \qquad (7)$$

where $\Phi$ is the sphericity of LDMPs and $\chi$ is the circularity of LDMPs. $\Phi$ is defined as the ratio of the surface area of an equivalent sphere and the surface area of LDMPs:

$$\Phi = A_{sph}/A_p \qquad (8)$$

where $A_{sph}$ is the surface area of the equivalent sphere (m$^2$), and $A_p$ is the surface area of LDMPs (m$^2$). $\chi$ is the circularity of LDMPs, defined as

$$\chi = P_{mp}/P_c \qquad (9)$$

where $P_{mp}$ is the maximum projection perimeter (m), and $P_c$ is the perimeter of a circle with the same area as the maximum projection area of the LDMPs (m). It is noted that for sphere-shaped LDMPs, $\Psi \approx 1$.

## The MICP model
Microbially induced calcium carbonate precipitation relies on biological processes, which are highly dependent on the presence of bacteria[28]. Kooi et al.[17] pointed out that the number and weight of bacteria are much less than algae. Thus, the present study only considers autotrophic MICP and calcite as modeling processes. The calcium flux ($J$; μmol cm$^{-2}$ s$^{-1}$) at the plastic surface is estimated using Fick's first law of diffusion[29]:

$$J = D_{Ca^{2+}} d[Ca_T]/dz \qquad (10)$$

where $d[Ca_T]/dz$ is the concentration gradient in the region adjacent to the plastic surface and $D_{Ca^{2+}}$ is the ionic diffusion coefficient for MICP.

This equation is functional under steady-state flux conditions since the concentration gradient mainly occurs within $500\,\mu m$ above the biofilm surface[25]. In a typical saturation state of 1.71, the precipitation rate of calcite is estimated as 5 μmol cm$^{-2}$ s$^{-1}$ [25]. The MICP rate is adopted under the assumption of the similar biofilm formation for buoyant plastics, which are polyethylene (PE) and polypropylene (PP). Due to their similar composition and structure, the biofilm formation conditions on most PE and PP surfaces are identical. However, the MICP rate cannot be directly adopted as a constant. A linear relationship between the light intensity and the MICP rate is established, which is highly related to photosynthetic strength. When the light availability decreases with increasing water depth, photosynthesis is weakened and the concentration gradient becomes insignificant, which terminates MICP.

$$P_{Ca} = I_z/I_0 {}^* J \qquad (11)$$

where $P_{Ca}$ is the MICP rate at depth z and $C_0^{Ca}$ is the MICP rate at the sea surface.

Oversaturated calcite in seawater cannot dissolve easily. With increasing depth, the increment of dissolved inorganic carbon content relative to total alkalinity enhances the solubility of calcite, leading to an unsaturated state[24]. The calcite saturation depth (CSD) is defined as the depth where the calcite unsaturation state first occurs[42]. Calcite on the LDMPs may partially dissolve when sinking under CSD and may regain buoyancy accordingly. However, CSD is often located in deep oceans where the seafloor is higher than CSD in many places. To study the impact of CSD, we introduce the calcite dissolution process when the adopted seawater condition shows the sinking depth is below the CSD[24].

## The biofouling growth model
**Biofouling formation in open oceans.** The biofouling formation on the material surface is a combination of microorganisms, including bacteria, algae, and/or fungi. For LDMPs, the biota types on the biofouling cover may be related to the original size, shape, and surface

roughness of LDMPs. The biofouling process, according to previous studies, can be described as follows[63,64]:

(1) Mass transfer of macromolecules to the surface and formation of an adsorbed layer, or EPS;
(2) Transport of microorganisms to the absorbed layer;
(3) Establishment of a strong binding between the microorganisms and surface layers;
(4) Cell metabolism, including reproduction, growth, and mortality.

To simplify the above-mentioned processes, biofilms comprise three main groups: Slime, non-shell organisms, and calcareous-type fouling[64]. The slime includes absorbed inorganic and organic matter, trapped silt and detritus, and other unidentified slimes. Non-shell organisms refer to most microorganisms and non-shell macroorganisms[64]. Calcareous fouling is a new type of modeled fouling, accompanied by biofouling, which is quantitated by the calcification model. These calcite precipitates are formed in a weakly alkaline environment created by the photosynthesis process and supersaturated calcium environment[25]. The formed $CaCO_3$ can attach tightly to plastic surfaces even after the mortality and decomposition of photosynthesized organisms.

**Modeling the biofouling process.** The biofouling on the material surface has been formulated, assuming all environmental conditions available for simulation and quantification. Equation (12) is used to evaluate the growth and thickness of biofouling attachment on the prospected targets[64,65].

$$BG = f_1(SST, psu, pH, \upsilon, I, S, t, m_t, \sigma, \theta_c, R_t) \tag{12}$$

where $BG$ is the growth rate of biofouling on a given surface, $SST$ is the seawater temperature, $psu$ is salinity (dissolved salt content of seawater), $pH$ is acidity, $\nu$ is the speed of the water flow, $I$ is light intensity, $S$ is the concentration of nutrients, $t$ is the time of the exposure to water, $m_t$ is the micro-texture of the surface, $\sigma$ is the surface potential, $\theta_c$ is the contact angle, which is a wettability measurement, and $R_t$ is the roughness parameter. Besides, surface color and contour also impact biofouling growth, although the relationship is not well-established[63].

The behavior of LDMPs upon biofouling is quite different from that of floating vessels. As biofouling grows on plastic surfaces, LDMPs accumulate weight and form a mixed batch of particles covered by calcite precipitates and biofilms[58]. Once LDMPs' density exceeds surrounding seawater density, settling begins, and the particles submerge below the sea surface[17]. Temperature, salinity, and seawater density vary with increasing depth, and the growth and bio-activity are gradually constrained by decreased light intensity and temperature[66]. Eventually, biofilms stop growing and even fade away. LDMPs may become suspended, float up, or sink depending on the surrounding seawater density.

Furthermore, Eq. (12) is just a conceptual model that includes all possible biofouling variables under marine conditions. However, this type of model can not be used for real-time predictions since validation of Eq. (12) with all parameters in the conceptual model requires enormous field measurements and laboratory tests, which are challenging to realize. We aim to evaluate the long-term settling of LDMPs in the marine environment, describing the vertical distribution of LDMPs in seawater. Thus, a simplified biofouling model is necessary to eliminate parameters and conditions that have little impact on the vertical settling of LDMPs.

The previous model for LDMPs' biofouling growth and density is presented below[17]

$$\rho_{tot}(z,t) = \frac{r_{pl}^3 \rho_p + \left[(r_{pl} + t_{bf})^3 - r_{pl}^3\right]\rho_{bf}}{(r_{pl} + t_{bf})^3} \tag{13}$$

where $\rho_{tot}(z, t)$ denotes the density of LDMPs with biofouling attachments at a given depth $z$ and time $t$ of seawater exposure; $r_{pl}$ is the radius of LDMPs (assuming a sphere shape); $t_{bf}$ is the average thickness of biofouling; and $\rho_{bf}$ represents the density of biofouling attachment. Based on Eq. (13), the impact of plastic shape is considered in the present model. LDMPs can be in various forms, and three basic shapes are commonly found in field measurements: fragment (near sphere), film, and fiber. For simplification, the film shape is standardized as a round shape with uniform thickness; the fiber shape is modeled as a cylinder with uniform lines and constant radius; and the fragment shape is modeled as a sphere. Thus, the $\rho tot(z,t)$ would be transformed as below:

$$\rho_{tot}(z,t) = \frac{\rho_p t_{pl} + 2t_{bf}\rho_{bf}}{t_{pl} + 2t_{bf}}(firm)(\text{Neglect edges}) \tag{14}$$

$$\rho_{tot}(z,t) = \frac{r_{pl}^2 \rho_p + \left[(r_{pl} + t_{bf})^2 - r_{pl}^2\right]\rho_{bf}}{(r_{pl} + t_{bf})^2}(fiber)(\text{Neglect both ends})$$

$$\tag{15}$$

$$\rho_{tot}(z,t) = \frac{r_{pl}^3 \rho_p + \left[(r_{pl} + t_{bf})^3 - r_{pl}^3\right]\rho_{bf}}{(r_{pl} + t_{bf})^3}(sphere) \tag{16}$$

where $t_{pl}$ represents the thickness of the plastic firm. The surface area is calculated using separate equations according to the given shapes.

**Film.** The thickness of biofouling ($t_{bf}$) on film is calculated by

$$t_{bf} = \frac{V_{tot}}{\pi r_{pl}^2} - t_{pl} \tag{17}$$

$$V_{tot} = V_{pl} + V_{bf} \tag{18}$$

$$V_{pl} = t_{pl}\pi r_{pl}^2 \tag{19}$$

$$S_{pl} = 2\pi r_{pl}^2 \tag{20}$$

where $V_{tot}$, $V_{pl}$, and $V_{bf}$ are the total, plastic, and biofouling volumes, respectively; $S_{pl}$ is the surface area of LDMPs; and $t_{pl}$ is the thickness of the plastic film. Here, only algae are considered the primary fouling objects since bacteria are much smaller than algae, and the number is twice as small as algae.

**Fiber.** The thickness of biofouling ($t_{bf}$) on fiber is calculated by

$$t_{bf} = \sqrt{\frac{V_{tot}}{\pi l_{pl}}} - r_{pl} \tag{21}$$

$$V_{tot} = V_{pl} + V_{bf} \tag{22}$$

$$V_{pl} = l_{pl}\pi r_{pl}^2 \qquad (23)$$

$$S_{pl} = 2\pi r_{pl} l_{pl} \qquad (24)$$

where $l_{pl}$ represents the fiber length, and $r_{pl}$ is the radius of the cross-section.

**Sphere.** The thickness of biofouling ($t_{bf}$) on a sphere is calculated by

$$t_{bf} = \sqrt[3]{V_{tot}\frac{3}{4\pi}} - r_{pl} \qquad (25)$$

$$V_{tot} = V_{pl} + V_{bf} \qquad (26)$$

$$V_{pl} = \frac{4}{3}\pi r_{pl}^3 \qquad (27)$$

$$S_{pl} = 4\pi r_{pl}^2 \qquad (28)$$

Here, only algae are considered as the primary fouling objects since bacteria are much smaller than algae, and the number is twice less than that of algae. Furthermore, the biomass of algae in global oceans is used as a proxy to scale the overall biofouling potential from microbes, phytoplankton, zooplankton, marine snow, and ingestion and inclusion in feces, widely accepted by other models[67]. $V_{bf}$ is represented as the sum of calcite precipitates and attached algae:

$$V_{bf} = (V_A A)S_{pl} + V_{minals} \qquad (29)$$

where $V_A$ represents the average volume of algae, $A$ represents the number of algae attached to the plastic surface; and $V_{minals}$ is the volume of attached calcite precipitates. To simplify the calculation, we assume that the attachment and algal growth are independent of MICP. Thus, the algal growth is calculated as[17]

$$\frac{dA}{dt} = \frac{\beta_A A_A}{\theta_{pl}} + \mu_A(T,I)A - m_A A - Q_{10}^{(T-20)/10}R_{20}A \qquad (30)$$

where four terms are included to present the fouling behavior of algae, namely collision[54], growth, morality, and respiration. The collision between algae and LDMPs is described by ambient algal concentration $A_A$ (no. m$^{-3}$) and encounter kernel rate (m$^3$ s$^{-1}$). Algal concentration is calculated through the chlorophyll-$a$ profile using the chlorophyll-$a$/carbon ratio and carbon/algal cell ratio, which depends on the temperature ($T$) and light-dependent ($I_z$) equation:

$$Chl\ a : C = 0.003 + 1.0154e^{0.050T}e^{-0.057I_z/10^6\mu'} \qquad (31)$$

where nutrients are assumed to be sufficient for algal growth.

A well-accepted Gaussian methodology is applied to quantify the vertical chlorophyll-$a$ profile[68]:

$$Chl(z) = \frac{Chl\ a(z)}{\overline{Chla_{Z_{base}}}} = C_b - sz + C_{max}e^{-((z-Z_{max})/\Delta z)^2} \qquad (32)$$

where $C_b$ is the normalized surface concentration; $s$ is the normalized slope; $C_{max}$ is the normalized maximum concentration; $Z_{max}$ represents the maximum concentration depth; $\Delta_z$ is the width of the peak concentration; and $\overline{Chla_{Z_{base}}}$ is the average chlorophyll-$a$ concentration of the vertical profile. Equation (32) is executed by a piecewise function with nine ranges based on the Chl $a$ surface concentration, which is acquired from NASA[52,68] (Table S3).

The collision between algae and LDMPs is calculated through the existing aggregation model for marine algal flocs[69]. The encounter kernel rate $\beta_A$ includes three compartments: the Brownian motion ($\beta_{ABrownian}$), differential settling ($\beta_{Asettling}$), and advective shear ($\beta_{AShear}$) collision frequencies. These different encounter kernel rates are expressed as

$$\beta_{ABrownian} = 4\pi(D_{pl} + D_A)(r_{tot} + r_A) \qquad (33)$$

$$\beta_{Asettling} = 0.5\pi r_{tot}^2 V_s \qquad (34)$$

$$\beta_{AShear} = 1.3\gamma(r_{tot} + r_A)^3 \qquad (35)$$

where $D_{pl}$ and $D_A$ are the diffusivities of LDMPs and algal cells, and $\gamma$ is the shear rate. For various shapes of LDMPs, $r_{tot}$ can not be directly calculated. Thus, the equivalent spherical diameter (ESD) is introduced to approximate $r_{tot}$ in Eqs. (33), (34), and (35)[26]:

$$r_{tot}^{ESD} = \sqrt[3]{6V_{tot}/\pi} \qquad (36)$$

Based on this conversion, the diffusivity of LDMPs and algae is calculated as follows:

$$D_{pl} = \frac{k(T + 273.16)}{6\pi\mu_{sw}r_{tot}^{ESD}} \qquad (37)$$

$$D_A = \frac{k(T + 273.16)}{6\pi\mu_{sw}r_A} \qquad (38)$$

where $k$ is the Boltzmann constant (m$^2$ kg s$^{-2}$ K$^{-1}$), $\mu_{sw}$ is the dynamic water viscosity (kg m$^{-1}$ s$^{-1}$). Based on Eqs. (33)–(38), the collision of algae with LDMPs can be calculated.

The algal growth (term 2 in Eq. (28)) is modeled as:

$$\mu(T_z, I_z) = \mu_{opt}(I_z)\Phi(T_z)\ for\ T_z \in [T_{min}, T_{max}] \qquad (39)$$

where $\mu_{opt}(I_z)$ is the algal growth rate under optimal temperature with a specific light intensity $I_z$, and $\Phi(T_z)$ is the temperature influence on growth rates[66].

$$\mu_{opt}(I_z) = \mu_{max}\frac{I_z}{I_z + \frac{\mu_{max}}{\alpha}\left(\frac{I_z}{I_{opt}} - 1\right)^2} \qquad (40)$$

$$\Phi(T_z) = \left[(T_z - T_{max})(T_z - T_{min})^2\right]/[T_{opt} - T_{min} \times ((T_{opt} - T_{min}) $$
$$(T_z - T_{opt}) - (T_{opt} - T_{max})(T_{opt} + T_{min} - 2T_z))] \qquad (41)$$

In Eq. (41), $I_z$ is the light intensity at depth $z$, $I_{opt}$ is the optimal light intensity for algal growth, $\mu_{max}$ is the maximum growth rate under optimal conditions, and $\alpha$ is the initial slope. In Eq. (42), $T_{max}$, $T_{min}$, and $T_{opt}$ are the maximum, minimum, and optimal temperatures to sustain algal growth. For temperatures outside the boundary of [$T_{min}$, $T_{max}$], $\mu_{max}$ is zero. The light intensity at a given depth $z$ is calculated according to the law of Lambert-Beer:

$$I_z = I_0 e^{\varepsilon z} \qquad (42)$$

where $I_0$ is the surface light intensity, and $\varepsilon$ is the extinction coefficient. Light availability at the sea surface is expressed using a sinusoidal function:

$$I_0 = I_m sin(2\pi t) \qquad (43)$$

 

where $I_m$ is the light intensity at noon. As $I_O$ never becomes negative, $I_O = 0$ when $\sin(2\pi t) < 0$. The light extinction is assumed to be dominated by water and algae-induced extinctions and can be expressed as:

$$\epsilon = \epsilon_w + \epsilon_p Chl\ a \tag{44}$$

where $\varepsilon_w$ and $\varepsilon_p$ are constant values to quantify water and chlorophyll-a extinction coefficients, respectively.

### Profiles of seawater salinity, temperature, and density

Seawater properties are critical for model simulation. Previous studies calculated vertical seawater salinity and temperature distribution using a fifth-order polynomial function against seawater depth. However, it is a rather mechanical and linear simulation that cannot reflect the reality of seawater at different locations. As a result, the RTOFS is introduced to acquire more practical and localized data to estimate seawater salinity and temperature from the surface to the seafloor under the National Centers for Environmental Prediction, which is based on an eddy-resolving 1/12° global hybrid coordinate[50,53]. The RTOFS provides seawater temperature and salinity nowcasting and forecasting the current date and next seven days' predictions. The data types assimilated include in situ temperature and salinity profiles from various sources and remotely sensed SST, SSH, and sea-ice concentrations. The operational ocean model configuration has 41 hybrid layers and a horizontal grid size of (4500 × 3298). The grid has an Arctic bi-polar patch north of 47°N and a Mercator projection south of 47°N through 78.6°S[53].

In the present study, only the nowcasting dataset is adopted. We have collected the nowcasting dataset from December 18, 2021, to December 18, 2022, using an auto-web crawler programmed by Python 3.7. Both salinity and temperature profiles include 33 available depths from the sea surface to the seafloor. For a continuous simulation, the salinity and temperature at a given depth $z$ are assumed to vary linearly within each layer:

$$T_z = \frac{T_{up} - T_{down}}{z_{up} - z_{down}} \times z + T_{up}, z \in [z_{down}, z_{up}] \tag{45}$$

$$S_z = \frac{S_{up} - S_{down}}{z_{up} - z_{down}} \times z + S_{up}, z \in [z_{down}, z_{up}] \tag{46}$$

For Eq. (45), $T_{up}$ and $T_{down}$ are estimated temperatures on the top and bottom of a given layer, respectively, and $z_{up}$ and $z_{down}$ are the top and bottom depths for that layer. For Eq. (46), $S_{up}$ and $S_{down}$ are the salinity estimations on the top and bottom of that layer, respectively.

**Seawater density.** According to Sharqawy et al.[70], the best-fit model with measured seawater density under atmosphere pressure is constructed by salinity and temperature:

$$\begin{aligned}\rho_{sw,z} &= a_1 + a_2 T_z + a_3 T_z^2 + a_4 T_z^3 + a_5 T_z^4 + b_1 S_z + b_2 S_z T_z + b_3 S_z T_z^2 \\ &\quad + b_4 S_z T_z^3 + b_5 S_z T_z^4\end{aligned} \tag{47}$$

$$a_1 = 9.999 \times 10^2, a_2 = 2.034 \times 10^{-2}, a_3 = -6.162 \times 10^{-3},$$

$$a_4 = 2.261 \times 10^{-5}, a_5 = -4.657 \times 10^{-8}, b_1 = 8.020 \times 10^2$$

$$b_2 = -2.001, b_3 = 1.667 \times 10^{-2}, b_4 = -3.060 \times 10^{-5}, b_5 = -1.613 \times 10^{-5}$$

where $a_1$–$a_5$ and $b_1$–$b_5$ are constant values, $T_z$ (°C) is the temperature at a given depth $z$ (m), and $S_z$ is the salinity at $z$ (m). The temperature and

salinity range for Eq. (35) are $0 < T < 180°C$ and $0 < S < 16\%$, respectively, and the accuracy errors are expected to remain within 1.5%.

**Seawater viscosity.** The seawater viscosity ($\mu_{sw}$, kg m⁻¹ s⁻¹) is also calculated by seawater temperature ($t$,°C), salinity ($S$, unitless), and pure water viscosity ($\mu_w$, kg m⁻¹ s⁻¹) given by IAPWS 2008 with an error of ±0.05%. The expression for the calculation is:

$$u_{sw} = u_w(1 + AS + BS^2) \tag{48}$$

where

$$\begin{aligned}A &= 1.541 + 1.998 \times 10^{-2} t - 9.52 \times 10^{-5} t^2 \\ B &= 7.974 - 7.561 \times 10^{-2} t + 4.724 \times 10^{-4} t^2 \\ u_w &= 4.2844 \times 10^{-5} + \left[0.157(t + 64.993)^2 - 91.296\right]^{-1}\end{aligned} \tag{49}$$

The accuracy error of Eq. (49) is expected to maintain within 1.5%.

### Simulation of observed LDMP distribution in open oceans

The Monte Carlo simulation methodology was introduced, and a group of simulated LDMPs was used to study the distribution patterns of LDMPs along the vertical direction based on available field investigations (Table S1). Because the exact number of LDMPs imported to the specific geographic location is almost impossible to acquire, the comparison mainly focused on the characteristics of LDMPs vertical settling patterns rather than the exact concentration distribution of LDMPs. To maximize the proximity to reality, the vertical seawater salinity and temperature profiles are acquired with latitude and longitude as the sampling location given by the available field measurements[13,14,48,49] through the global real-time ocean forecasting system (RTOFS)[53,56,65].

### Overall model structure and verification

The model program, as described in a general flowchart (Fig. S3), is composed of four basic modules corresponding to four critical processes regarding the sinking process of LDMPs. The red module describes the surface changes of LDMPs under MICP and biofouling. The blue module represents designated seawater conditions given a specific geographic location and depth. The green module presents light intensity variation for photosynthesis. The purple module represents the position and velocity of given LDMPs under calculation based on given conditions, such as seawater conditions, LDMP properties and postures, MICP, biofilm growth, and so on. After settling the initial configuration, the model utilizes a Markov Chain procedure to calculate the trajectory of all LDMPs along a continuous time interval. The final results include the trajectories of randomly generated LDMPs, whose characteristics matched with observations.

## Data availability

All source data and generated data needed to evaluate the conclusions in the paper have been deposited at: https://doi.org/10.5281/zenodo.11262785 or the Supplementary Materials. Materials can be obtained by pending scientific review and a completed material transfer agreement. Requests for materials should be submitted to eddyzeng@jnu.edu.cn.

## Code availability

The original code of the LDMPs vertical settling model, development datasets, simulation results, and figure files are available online at: https://doi.org/10.5281/zenodo.11262785.

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

## Acknowledgements

This work was supported by the Southern Marine Science and Engineering Guangdong Laboratory (Zhuhai) (No.SML2021SP208), the Guangdong Basic and Applied Basic Research Foundation, China (No. 2020A1515110888), and the National Natural Science Foundation of China (No. 42007313).

## Author contributions

E.Y.Z. initiated the project. E.Y.Z. and X.S. conceived and designed the study. M.X. and L.M. collected all the data. X.S., Y.Z., M.X., and E.Y.Z. contributed to the model configuration and data analyzes. X.S., Y.Z., and E.Y.Z. wrote the manuscript. All authors contributed to the data interpretation and manuscript editing.

## Competing interests

The authors declare no competing interests.
