## [Peer Review File · Nature Communications]

Calcite carbonate sinks low-density plastic debris in open oceansReviewers' comments:

Reviewer #1 (Remarks to the Author):

By Reviewer Michiel Van Melkebeke:

The authors of the manuscript titled “Sinking the unsinkable: Ballast effects of calcite on low-density plastic debris in open oceans” have provided a sensible and well-structured rebuttal. The article itself is carefully written with a strong and fitting title. The manuscript proposes a new model to describe the sinking behavior of low-density plastic debris in the marine environment by explicitly accounting for calcite precipitation. The authors claim that, next to biofouling, calcite precipitation greatly impacts the sinking pattern of plastic debris.

The presented results are interesting and particularly relevant as the discrepancy between the observed and the estimated amount of plastic debris on the oceanic surface is still only partially understood. Figure 3 in particular provides an excellent graphical summary of the simulations, where a comparison is made between the results of the model with and without accounting for calcite precipitation. From this, the improvement of the newly presented model in predicting the vertical trajectory of plastic debris in oceanic waters appears evident. However, much of their work is based on the assumption that calcite precipitates are much more difficult to detach from the plastic surface than biofilms. While this does seem reasonable, additional confirmation or validation aside from the single reference made originating from 1996 by the efforts of Hartley et al. would definitely strengthen their work. In addition, the authors mention the inconsistency of field measurements with the first model that accounted for biofouling on plastic surfaces proposed by Kooi et al. in 2017, which they in fact also employed as a basis to account for biofouling in their new model. There are, however, various improvements made on this first model such as the one proposed by Fischer et al. (2022) in their work titled “Modelling submerged biofouled microplastics and their vertical trajectories” and related efforts put forward by Lobelle et al. (2021) in the article titled “Global Modeled Sinking Characteristics of Biofouled Microplastic”.

Nevertheless, I find the explicit inclusion of calcite precipitation in a model to describe the vertical trajectory of plastic particles in a marine environment intriguing. The potential significance of the effect of calcite precipitation on the sinking behavior of plastic debris in oceanic waters could greatly affect the marine ecosystem and its global carbon cycle beyond the current understanding.

Finally, in general, the described methodology is satisfactory and the discussion highlights some interesting insights. Yet, I do have some specific comments:

Line 20-21: I would suggest to modify: “It further reveals that plastic debris can distribute at all depths after years of drifting with two accumulation zones at 60-400 m of the water column and *at the seafloor*”

Line 86-87: Missing word: “The vertical seawater density *variation* is a critical factor during plastic settling (...)”

Line 123-124: Reformulation: “But calcite precipitates are more difficult to *detach from* plastic surfaces than algae, as they are embedded within EPS”

Line 128-129: Correction: “Some plastic debris (e.g. 20 μm film) settle slowly and steadily, *whose* density can grow much higher than seawater density, overcoming the high viscosity caused by large specific surface area.”

Line 208-209: Correction: “But the dispersion of *this* plastic debris was much quicker than those in the size range of 0-200 μm .”

Line 227: Figure 5: It is confusing that the legend in the top right corner uses different colors to label the shapes of the plastic particles, while only the point shape is in fact relevant for this description.

Line 237-238: Correction: “In the first 100 days, the vast majority of plastic debris *remains at* the sea surface to a depth of 200 m, while a few large-sized plastic debris (> 400 μm) have begun to settle into the deep water column (Figs. 5A-5B).”

Line 251-252: This statement is unclear or wrong. I advise the authors to clarify or rectify: “Our simulation results also indicate that small-sized plastic debris have entered the upper 100-m water column in less than 300 days.”

Line 342 and 344: What does “angle only” mean? Perhaps the authors meant to write “*algae* only”? If so, please correct.

Line 431: Correction: “*reparameterizations*”

Reviewer #1 (Remarks on code availability):

NA

Reviewer #2 (Remarks to the Author):

Review report for Sun et al., “Sinking the unsinkable: Ballast effects of calcite on low-density plastic debris in open oceans”

This article presents simulation results of the impacts of a previously overlooked phenomenon: calcite precipitation on plastic surface. The authors present simulation results showing that calcite precipitation may explain a large part of the sinking of positively buoyant plastics. They attempt an evaluation of their model results by comparing vertical profiles of measured plastic debris with their simulation results. Then, the authors compare the simulated impacts of calcite precipitation on various plastics shapes,

sizes and densities and conclude by affirming that calcite precipitation explains the “missing plastic” puzzle in the surface ocean.

This article studies an overlooked process that could significantly impact plastic distribution in the global ocean. While the question of whether calcification is a significant process to plastic’s fate in the ocean is interesting and the methodology employed (numerical modeling) is adapted to the question, the authors fail to justify in the article the importance of such study. Overall, the article lacks references to justify that calcite precipitation may be an important phenomenon, or that it has been observed at all on plastic debris. The methods are also poorly described. It is not even clear whether the model presented here was entirely designed for the study, or if it’s based on previous work. There is no reference for the data used to force the model (physical and biogeochemical conditions). Overall, there are very few references in the manuscript and entire paragraphs are given without referencing any article, or even some of this study’s results. Therefore, the whole article reads as a thought experiment and remains too theoretical to recommend publication in the current state.

I am detailing below some of my major comments to improve this manuscript.

Major comments:

In the introduction, a paragraph is missing to explain why calcite precipitation is important for plastic debris distribution. References are needed here to justify that this phenomenon is worth investigating and exploring in a model. This would justify the present study.

Many sentences lack references, as a result, a lot of the arguments presented remain pure speculation.

For instance, lines 89-94 “The distribution of plastic debris within the second accumulation zone is a dynamic interplay, intricately influenced by multifarious variables such as incident light intensity, nutrient level, prevailing seawater condition, distance from emission sources, and the distinctive attributes of said sources. Consequently, one may observe a spectrum of concentrations ranging from several to hundreds of plastic particles per cubic meter.” But also lines 109-115, 190-200.

One thing I found particularly confusing is that the dimensions of the plastic debris of interest in this study are not described until line 118. As a result, we do not know what kind of plastic debris are simulated and measured.

Lines 75-80: what data are you referring to? You also state that “The fitting performance indicates that improvements are achieved in simulating the vertical settling dynamics of plastic debris in open oceans.” compared to what? How can you affirm that the vertical dynamics are improved ?

In general, the figures could be improved. Many lack some units or information to help interpret their results. Moreover, the text describing the figures presents things that we can’t see on them. For instance lines 103-104: “Once the density of plastic debris surpasses the seawater density and overcome the seawater viscosity, plastic debris can settle on the seafloor (Fig. 1)” There is no mention of seawater density or viscosity in the figure.

Lines 177-179 (describing Fig.4): “This trend reveals that the settling of plastic debris is increasingly dependent on calcite precipitation and biofilm growth as the equivalent diameter of plastic debris becomes smaller due to photolysis and abrasion.” No mention of photolysis and abrasion anywhere in the figure (or the MS).

Figure 1: What's "ss" on the plots? How were the curves calculated? They seem far from the dots at some sites (e.g. North Pacific Ocean). No units on the x axis.

Figure 4. What's the shape category of the particles in A,B,C,G,H and I. How does this choice affect the results?

Figure 5 must be modified. As presented, we cannot distinguish the 3 shapes. What's the density of the particles? What's the simulation conditions?

Lines 184-189: "Moreover, sphere-shaped plastic debris are most likely to accumulate on the seafloor. Hence fibers and films are more abundant than spheres in the water column. A close examination of plastic debris sized below 1 mm indicates that fibers and films need much longer time than spheres to reach the seafloor, especially for those with equivalent diameters smaller than 100 μm , even though they take a shorter time to submerge into the water column." While this may be shown in your simulations, such results are not new and where seen by other authors. References and discussion of your results with regards to these previous studies is lacking here.

Line 253: "In summary, large plastic debris (> 200 μm) are concentrated at 60–400 m and on the seafloor" This conclusion (which is a major result of the study since it's also highlighted in the abstract) seems supported by none of the figures presented. Maybe in Figure 4 which presents results from theoretical simulations.

The sentence line 258-261 "However, such depletion of plastic debris in the upper 100 m water column can hardly be field-measured, because increasing plastic production and insufficient plastic waste management constantly supplement plastic debris to the sea surface." is awkward. It seems to justify that the results of this study may not be supported by data.

Line 265: " Since it requires a considerable amount of time for plastic debris to build up calcite precipitates and biofilms" This sentence lacks a reference, and is not true. Biofilms form within minutes to hours (see Paul-Pont et al., 2018)

Paul-Pont I, Tallec K, Gonzalez-Fernandez C, Lambert C, Vincent D, Mazurais D, Zambonino-Infante J-L, Brotons G, Lagarde F, Fabioux C, Soudant P and Huvet A (2018) Constraints and Priorities for Conducting Experimental Exposures of Marine Organisms to Microplastics. *Front. Mar. Sci.* 5:252. doi: 10.3389/fmars.2018.00252

Lines 267-273: "By analyzing the vertical distribution patterns of plastic debris, we can diagnose possible emission sources and their geographic locations (Fig. S3). If plastic debris collected at a given geographic location originate from nearby emission sources, the surface concentration of plastic debris would be higher than that resulting from distant emission sources. If plastic debris are derived from multiple emission sources with numerous distances, their vertical distribution patterns would contain multiple overlapping peaks. " Again, no reference for these affirmations. Moreover, these affirmations completely ignore the fact that ocean currents may also be vertical! Vertical currents have been shown to significantly influence plastic distribution in 3D (Richon et al., 2022, Mountford et al., 2019, Huck et al., 2022). Also, these sentences overlook the potential influence of fish and other animals on plastic vertical distribution through ingestion-migration-excretion (Justino et al., 2020).

Justino, A. K. S., Ferreira, G. V. B., Schmidt, N., Eduardo, L. N., Fauvelle, V., Lenoble, V., Sempéré, R., Panagiotopoulos, C., Mincarone, M. M., Frédou, T., & Lucena-Frédou, F. (2022). The role of mesopelagic fishes as microplastics vectors across the deep-sea layers from the Southwestern Tropical Atlantic. *Environmental pollution* (Barking, Essex : 1987), 300, 118988.

<https://doi.org/10.1016/j.envpol.2022.118988>

Richon C, Gorgues T, Paul-Pont I and Maes C (2022) Zooplankton exposure to microplastics at global scale: Influence of vertical distribution and seasonality. *Front. Mar. Sci.* 9:947309. doi: 10.3389/fmars.2022.947309

Huck T, Bajon R, Grima N, Portela E, Molines J-M and Penduff T (2022) Three-Dimensional Dispersion of Neutral “Plastic” Particles in a Global Ocean Model. *Front. Anal. Sci.* 2:868515. doi:

10.3389/frans.2022.868515

Mountford, A. S., & Morales Maqueda, M. A. (2019). Eulerian Modeling of the Three-Dimensional Distribution of Seven Popular Microplastic Types in the Global Ocean. *Journal of Geophysical Research: Oceans*, 124, 8558–8573. <https://doi.org/10.1029/2019JC015050>

One major question that I have is if biofilm detaches when plastic sinks, why would calcite precipitates remain on plastic surface? This is not justified anywhere in the manuscript.

Methods

In general, the methods (except for the model description and equations) lack important information. There is no information on whether the model used in this study was built specifically, or if it existed before. Again, there is no reference to calcite precipitation being a significant process (lines 309-311 have no reference).

Line 364-365: “Calcite precipitates are much denser than seawater and are difficult to detach, thus they are much more efficient than algae in increasing the density of plastic debris.” Needs a reference. Is there any study showing calcite precipitation on plastic debris?

Lines 387-406: what location did you use? How did you select the sampling sites?

Line 395-397: “We may move the coordinate of simulated seawater profiles a little bit from the original position if the depth of the RTOFS dataset at the original site cannot approach the measured depth”

Since we have no idea of the locations you are referring to, this sentence makes no sense.

The informations of size and density of the plastics should also be mentioned in the main manuscript.

Lines 405-406: “The floating plastic debris which are denser than the seawater may be caused by trapped air, hydrophobicity, and surface tension.” So, you simulated bubble formation?

Lines 409-420: Again, no mention of the observations, what observations? Where?, Who sampled? How was it sampled? I don’t understand the organisation of the model with colored modules.

Lines 421-422: “The vertical distribution patterns of plastic debris are estimated via the interpolation method to create smooth curves for comparison.” Comparison with what? What interpolation method?

The most problematic with the methods section is that there is no mention of how the simulations were performed. How long did they run for? Is that 0D, 1D, 2D, 3D? What data is used for forcing? Is there a control simulation? Did you simulate all processes separately? Did you simulate biofouling & calcite precipitation? Did you perform a simulation without calcite precipitation for comparison? As presented, it seems not. Thus, there is no way to know whether calcite precipitation is the process

actually impacting plastic export.

Minor corrections:

- Line 50: "Indian Ocean"
- Line 146: What's the "plastic debris simulator"?
- Lines 191-193: "Because spheres have the lowest specific surface areas among all shape categories, calcite precipitates and biofilms are much thicker on spheres than on fibers and films when similar overall density is achieved." Needs a reference

Reviewer #3 (Remarks to the Author):

Review of "Sinking the unsinkable: Ballast effects of calcite on low-density plastic debris in open oceans"

In this study, the authors develop a new model of explicit calcite biofouling of microplastics to investigate the transport potential of calcite ballast for moving plastic particles into the deeper ocean and away from the sea surface. They find that the combination of organic and calcite biofouling transports microplastics into the deeper ocean levels and leads to the accumulation of debris in the mesopelagic and near the seafloor. Deeper transport of microplastics is more consistent with observations. Their implementation of multiple plastic particle characteristics in addition to organic and calcite biofouling makes this a novel study that could be interesting to a broad audience. However, some of the arguments put forth are not fully convincing and more information is needed. Furthermore, the organization of the manuscript could be improved for readability. Starting the results section with the Monte Carlo simulations (from L141). Move Fig. 3 to Fig. 1 and remove panel A (or, consider moving Fig. 3 to the supplement and adding non-calcite ballast to current Fig. 2). Make current Fig. 1 new Fig. 2. Add density and chlorophyll profiles to that figure panels. Remove the red curve, add in the no-calcite profile to clearly demonstrate the impact. Move current Fig. 4 up to new Fig. 3. Rearrange text accordingly. I think these changes could improve the flow of the Results section and make it more understandable.

General comments:

Introduction: The size range of plastic debris should be introduced in the first paragraphs of the manuscript, to make it clear which plastics are being considered. This is also true for the discussion of the Kooi study (from L39), so that the reader understands this is a relevant study to the present topic. Also the paragraph from L44, explicitly stating the size ranges of the observed plastics to make it clear these are fairly compared to the Kooi oscillation mechanism.

Model validation section:

Figure 1: It would be very helpful to see an annual average chlorophyll map in Fig. 1. This could replace the current map, with the observational points still overlaid. Please remove the red lines from Figure 1 as they do not add any insight into the model assessment. Are the observations down the water column taken at the same time as the surface observations? Please also define the size range of the plastic

debris used in this model-obs comparison. And, are the units particles per m^{-3} ?

The Fréchet distance is not terribly convincing, as the observational curves are not convincingly similar to the observations themselves in many locations. I would prefer a more direct/less derived comparison method.

Paragraph starting at L81 is not well justified by the figures. Clarity could be improved by moving Fig. 4 to Fig. 3, which then demonstrates the 2 major accumulation zones for different particle types in context. If chlorophyll is added to Fig. 1 then this would help to demonstrate how biological activity is shaping water column plastic distributions. Why not also add density profiles to the panels of Fig. 1, and then you can remove Fig. S1A and also directly see how the density gradient is affecting accumulation zones? Fig. S1B is not very useful and can be removed.

Last 2 sentences from L89 “The distribution of plastic debris...” are vague and not demonstrated clearly from looking at the figures. Fig. 1 appears to show typically lower concentrations.

Paragraph starting from L95 is also not completely convincing. Particularly, L98 “No second accumulation zone...”. What about seasonal deep mixing (e.g. Fischer et al. 2022), and the strongly seasonal biomass? Could seasonality be determining the homogeneous distribution? And from the next sentence (L99) Kvale et al. (2020) <https://www.nature.com/articles/s41598-020-72898-4> Supplement has a sensitivity analysis showing that surface concentrations in the Arctic are highly sensitive to the buoyant fraction advecting from lower latitudes, where particles biofouled and in the subsurface below 40N tend not to escape to the Arctic. Can a figure strengthen the arguments put forth in this paragraph?

Paragraph from L109: Reference to a figure would help here. Better still would be overlaying density and chlorophyll on the panels of Fig. 1.

L160&L281: “It seems that...” this sentence is too conclusive, as other mechanisms are not tested, for example marine snow aggregation and fecal pellet sinking. These other mechanisms have been simply parameterized in another model and have similar results (mesopelagic accumulation), e.g. Kvale et al. (2020) <https://www.nature.com/articles/s41598-020-72898-4>. Likewise, biofouling was demonstrated to be able to distribute plastic particles deep into the water column (hundreds of meters) in slow oscillations by Fischer et al. (2022) <https://doi.org/10.5194/bg-19-2211-2022>. Can you explain why your biofouling model does not behave similarly to that of Fischer et al.?

Paragraph from L180. I really like this paragraph and associated Fig. 4. These are a highlight of the paper.

Paragraph from L219. Are there observations that can be used to validate these simulated patterns? Perhaps Zhao et al. (2022) <https://doi.org/10.1111/gcb.16089>.

Discussions: Kvale et al. (2020) <https://www.nature.com/articles/s41598-020-72898-4> Supplement had a whole sensitivity test (Figs. S6, S9) that demonstrated the role of biologically mediated (fecal pellet and marine snow aggregates) in sinking microplastic particles into the mesopelagic and in accumulation of plastic debris at these depths, but this is never discussed. Likewise, comparison to the Fischer et al.

(2022) <https://doi.org/10.5194/bg-19-2211-2022> study and the Lobelle et al. (2021) <https://doi.org/10.1029/2020JC017098> study is not done.

Supersaturated dissolution of calcite is an observed phenomenon and responsible for the regulation of deep ocean calcite fluxes, e.g. Subhas et al. (2022) <https://doi.org/10.1029/2022GB007388>. Choosing to omit this process should be discussed.

Specific comments:

L61: From “Ingestion by seabirds...” This sentence can be deleted as it is not obviously relevant.

L77: What is Fréchet similarity (please define)

L78: “The fitting performance...” this sentence is not justified by a comparison to a model without calcite ballast. Could model simulations that do not include calcite ballast be added to Fig. 1 panels?

L106: are ageing and fragmentation represented in this model? If not, please remove this sentence.

L122: This sentence should be rephrased for clarity. I think ‘resumption’ is the wrong word. And is in situ CO₂ affecting calcification rate? No dissolution is included in this model- correct? (please state explicitly) A sentence added around L122 to describe how calcite precipitates are modelled to come off the plastic particle would be helpful for understanding this section.

Fig. 2: Adding biofouling only (no calcite) trajectories, velocities, and densities to these panels would enhance understanding of how calcite is uniquely affecting the results.

L152: “Due to low light intensity...” the dominant control of light intensity on deep particle sinking is not clearly demonstrated by the results. Rephrasing to a density difference focus would be more clear.

L176: “According to the simulation results...” this sentence does not make sense to me because the initial density cannot be modified by the settling time. Please revise.

Next sentence (L177) I am confused here because up to now I thought the plastic particles are not losing mass. Please explicitly state this somewhere before this point. Also L193, plastic is warped in the model by biofilm? I am confused.

L214: “It seems that...” this sentence is self-contradictory. Could it be “maintain sinking” rather than “floating”? Or “small” should be “large”?

L285: “If plastic debris...” Do you have a citation for this statement?

Reviewer #3 (Remarks on code availability):

The code is insufficiently provided and with no documentation. I do not see how I would be able to reproduce the modelling from what is provided. As far as I can tell, only model outputs are provided.

Reviewer 1:

The authors of the manuscript titled “Sinking the unsinkable: Ballast effects of calcite on low-density plastic debris in open oceans” have provided a sensible and well-structured rebuttal. The article itself is carefully written with a strong and fitting title. The manuscript proposes a new model to describe the sinking behavior of low-density plastic debris in the marine environment by explicitly accounting for calcite precipitation. The authors claim that, next to biofouling, calcite precipitation greatly impacts the sinking pattern of plastic debris.

The presented results are interesting and particularly relevant as the discrepancy between the observed and the estimated amount of plastic debris on the oceanic surface is still only partially understood. Figure 3 in particular provides an excellent graphical summary of the simulations, where a comparison is made between the results of the model with and without accounting for calcite precipitation. From this, the improvement of the newly presented model in predicting the vertical trajectory of plastic debris in oceanic waters appears evident.

However, much of their work is based on the assumption that calcite precipitates are much more difficult to detach from the plastic surface than biofilms. While this does seem reasonable, additional confirmation or validation aside from the single reference made originating from 1996 by the efforts of Hartley et al. would definitely strengthen their work.

Response/action: We thank your recognition of our work. In the revised manuscript, extra references and evidence from material science are added to strengthen the statement that calcite precipitates are much more difficult to detach from the plastic surface than biofilms.

"Microbially induced calcium carbonate precipitation (MICP) is commonly observed under calcium-rich and high pH environments, such as ocean surface and soil matrix³²⁻³⁴. The present study considers MICP accompanied by biofouling (autotrophic path)³⁵. Most algae have negative surface charges³⁶, and continue attracting over-saturated Ca^{2+} in the upper seawater column^{36,37}. Some biological processes, such as photosynthesis and hydrolysis of urea, can increase pH levels in surrounding water³³, triggering CaCO_3 to precipitate near the biofilm and become part of the biofilm exoskeleton^{32,38}. Different types of CaCO_3 , including calcite and aragonite³⁹, can be produced by MICP. However, aragonite formation requires temperatures over 30 °C⁴⁰. Thus, calcite is considered the primary precipitate on LDMPs⁴¹. With a density over 2.63 g cm⁻³, calcite could be an adequate ballast to sink LDMPs to ocean sediments." (lines 71–80)

"Moreover, calcite precipitates are difficult to come off plastic surfaces³⁴. Laboratory studies revealed that the binding between calcite and LDMPs is so strong that calcite can be used as a coat to strengthen plastics in cementitious materials^{37,50,51}." (lines 100–103)

"The present model also considers the impact of calcite dissolution after LDMPs penetrate the calcite saturation depth (CSD)³³. The degree of saturation of the seawater with respect to calcite is the driving force behind its precipitation or dissolution⁵⁶. The level of calcite saturation decreases in the cold deep water. Because the solubility of calcite increases with increasing seawater depth/pressure, and the concentration of dissolved inorganic carbon

increases relative to total alkalinity levels in the deep ocean⁵⁷. Consequently, our model simulation indicates that all LDMPs suffer slight calcite loss and settling velocity deceleration after passing the CSD. However, the impact of calcite dissolution is relatively small for two reasons. First, the settling velocity of LDMPs is relatively high. Most LDMPs can reach the sea floor within less than one year, leading to insufficient dissolution. Second, the CSD varies regionally from 1,000 m to 4,600 m³³. Calcite dissolution may never occur when the seafloor is above the CSD. For instance, the CSD in most regions of the Atlantic Ocean is below 3,500 m³³. In comparison, the average depth of the Atlantic Ocean is 3,646 m." (lines 122–134)

In addition, the authors mention the inconsistency of field measurements with the first model that accounted for biofouling on plastic surfaces proposed by Kooi et al. in 2017, which they in fact also employed as a basis to account for biofouling in their new model. There are, however, various improvements made on this first model such as the one proposed by Fischer et al. (2022) in their work titled “Modelling submerged biofouled microplastics and their vertical trajectories” and related efforts put forward by Lobelle et al. (2021) in the article titled “Global Modeled Sinking Characteristics of Biofouled Microplastic”.

Response/action: The reviewer's concerns are well taken. In the revised manuscript, brief discussions regarding the recent studies of Fischer et al. (2022) and Lobelle et al. (2021) are added. We highlight the high concentration of low-density microplastics (LDMPs) present in the ocean sediment, which can be properly explained by the present model and is beyond the capability of Fischer et al. (2022).

"However, the biofouling model encounters enormous challenges to explain the enrichment of LDMPs in the oceanic sediment. Modeled LDMPs were impossible to approach the ocean floor due to the continuing loss of biofilm²⁶. Fischer et al.²⁵ evaluated LDMP settling under another biofilm dynamic, assuming frustule attachment on LDMPs after settling breathe the twilight zone. Ingestion by marine organisms (fecal pellets) merely sinks a negligible fraction (0.13–0.19%) of LDMPs^{27,28}. Aggregation with biogenic particles (marine snow) could sink another 0.06–8.8% of LDMPs^{29,30}. Lobelle et al.³¹ and Fischer et al.²⁵ incorporated oceanographic processes with a settling model, including large-scale 3D advection, small-scale vertical turbulence, dynamic grazing, and wind-induced mixing, etc. The updated model allowed LDMPs to sink below the euphotic zone and mixed layer; however, only 15 of 10,000 LDMPs reached the ocean floor (>5,000 m) under favorable conditions²⁵." (lines 60–70)

"Depending on the settling conditions, most LDMPs could sink to the seafloor within tens to hundreds of days (Fig. 1B). Microbially induced calcium carbonate precipitation may serve as one of the critical factors in depleting LDMP concentrations at the ocean subsurface and enhancing LDMP accumulation at the oceanic sediment¹⁶." (lines 118–121)

Nevertheless, I find the explicit inclusion of calcite precipitation in a model to describe the vertical trajectory of plastic particles in a marine environment intriguing. The potential significance of the effect of calcite precipitation on the sinking behavior of plastic debris in

oceanic waters could greatly affect the marine ecosystem and its global carbon cycle beyond the current understanding.

Response/action: The reviewer's concerns are well taken. We completely agree with you regarding this point. We have noticed the potential impacts of calcite precipitation on plastic particles on the marine ecosystem and its global carbon cycle. Our next step is to study these impacts based on a new three-dimensional global model, which will add oceanographic processes, marine snows, and aggregation effects. The global carbon cycle will surely be one of our primary concerns due to extra carbon removal through plastic settling.

"The simulation results suggest that the seafloor is likely the ultimate destiny of LDMPs. If LDMPs with calcite precipitates eventually sink to the seafloor, it might impact the ocean benthic ecosystem, global carbon cycle, and ocean calcite cycle. All these potential risks require adequate assessments and systematic studies." (lines 327–331)

Finally, in general, the described methodology is satisfactory and the discussion highlights some interesting insights. Yet, I do have some specific comments:

Line 20-21: I would suggest to modify: "It further reveals that plastic debris can distribute at all depths after years of drifting with two accumulation zones at 60-400 m of the water column and *at the seafloor*"

Response/action: The reviewer's concerns are well taken. In the revised manuscript, the statement has been revised as suggested and double-checked throughout the entire manuscript.

"In oceanic sediment, abundant LDMPs were found at the Southwest Indian Ocean (900–1,000 m)¹², the Northeast Atlantic Ocean (1,400–2,200 m)¹², the Arctic Ocean (2,783–5,570 m)¹³, and the western Pacific Ocean (5,108–10,908 m) at the Mariana Trench¹⁴." (lines 48–51)

"...but rational explanations for LDMP vertical settling remain challenging." (lines 52–53)

"However, the biofouling model encounters enormous challenges to explain the enrichment of LDMPs in the oceanic sediment." (lines 60–61)

"With proper vertical drag coefficient, the LDMP settling would neither follow an oscillation pattern nor sink quickly to the ocean floor." (lines 164–165)

"The second accumulation zone (seafloor zone) is at the oceanic sediment." (line 260)

Line 86-87: Missing word: "The vertical seawater density *variation* is a critical factor during plastic settling (...)"

Response/action: The reviewer's points are well taken. In the revised manuscript, the statement has been revised as suggested and double-checked throughout the entire manuscript.

"Depending on how much calcite precipitates remain and how the seawater density varies,..." (line 137)

Line 123-124: Reformulation: "But calcite precipitates are more difficult to *detach from* plastic surfaces than algae, as they are embedded within EPS"

Response/action: The reviewer's concerns are well taken. In the revised manuscript, the statement has been revised as suggested and double-checked throughout the entire manuscript.

"...calcite precipitates are more difficult to detach from the plastic surface than biofilms." (lines 115–116)

Line 128-129: Correction: "Some plastic debris (e.g., 20 μm film) settle slowly and steadily, *whose* density can grow much higher than seawater density, overcoming the high viscosity caused by large specific surface area."

Response/action: The reviewer's concerns are well taken. In the revised manuscript, the statement has been revised for better understanding.

"Thus, LDMPs can retain most of the gained density and continue to settle until the negative buoyancy is balanced with the vertical drag." (lines 116–118)

Line 208-209: Correction: "But the dispersion of *this* plastic debris was much quicker than those in the size range of 0-200 μm ."

Response/action: The reviewer's concerns are well taken. In the revised manuscript, the statement has been revised as suggested and double-checked throughout the entire manuscript.

"The maximum sizes for these sinking fibers and films can be one order of magnitude larger than those for sinking spheres at equivalent diameters (equivalent in volume to sphere diameter)." (lines 203–205)

Line 227: Figure 5: It is confusing that the legend in the top right corner uses different colors to label the shapes of the plastic particles, while only the point shape is in fact relevant for this description.

Response/action: The reviewer's concerns are well taken. The different colors are utilized to represent LDMP sizes rather than shapes. The shape is represented by the shape of the point. In the revised manuscript, Fig. 5 is updated to Fig. 6, with separate plots to better exhibit the shape of LDMPs.

Line 237-238: Correction: "In the first 100 days, the vast majority of plastic debris *remains* at the sea surface to a depth of 200 m, while a few large-sized plastic debris (> 400 μm) have begun to settle into the deep water column (Figs. 5A-5B)."

Response/action: The reviewer's concerns are well taken. In the revised manuscript, the statement has been revised as suggested.

"In the first 20th days, the vast majority of LDMPs remain at the sea surface to a depth of 1,500 m, while a few LDMPs have begun to settle into the deep water column (Fig. 6A). " (lines 300–301)

Line 251-252: This statement is unclear or wrong. I advise the authors to clarify or rectify: "Our simulation results also indicate that small-sized plastic debris have entered the upper 100-m water column in less than 300 days."

Response/action: The reviewer's concerns are well taken. In the revised manuscript, the statement has been revised as suggested.

"As the settling time passes the 100th day, most LDMPs begin to diffuse into the deep water column, and the subsurface accumulation zone begins to fade away due to lack of continuous supply of LDMPs (Fig. 6B)." (lines 304–306)

Line 342 and 344: What does "angle only" mean? Perhaps the authors meant to write "*algae* only"? If so, please correct.

Response/action: The reviewer's concerns are well taken. In the revised manuscript, the statement has been corrected as suggested.

"... $m_{bf}(t,z)$ is the mass of biofilms (algae only, kg),..." (line 397)

"... $V_{bf}(t,z)$ is the volume of biofilms (algae only, m^3),..." (lines 398–399)

Line 431: Correction: "*reparameterizations*"

Response/action: The reviewer's concerns are well taken. Yet, since we have no longer utilize the Fréchet distance. The statement has been removed in the revised manuscript.

Reviewer #1 (Remarks on code availability):

N/A

Reviewer 2:

Review report for Sun et al., “Sinking the unsinkable: Ballast effects of calcite on low-density plastic debris in open oceans”

This article presents simulation results of the impacts of a previously overlooked phenomenon: calcite precipitation on plastic surface. The authors present simulation results showing that calcite precipitation may explain a large part of the sinking of positively buoyant plastics. They attempt an evaluation of their model results by comparing vertical profiles of measured plastic debris with their simulation results. Then, the authors compare the simulated impacts of calcite precipitation on various plastics shapes, sizes and densities and conclude by affirming that calcite precipitation explains the “missing plastic” puzzle in the surface ocean.

This article studies an overlooked process that could significantly impact plastic distribution in the global ocean. While the question of whether calcification is a significant process to plastic’s fate in the ocean is interesting and the methodology employed (numerical modeling) is adapted to the question, the authors fail to justify in the article the importance of such study. Overall, the article lacks references to justify that calcite precipitation may be an important phenomenon, or that it has been observed at all on plastic debris. The methods are also poorly described. It is not even clear whether the model presented here was entirely designed for the study, or if it’s based on previous work. There is no reference for the data used to force the model (physical and biogeochemical conditions). Overall, there are very few references in the manuscript and entire paragraphs are given without referencing any article, or even some of this study’s results. Therefore, the whole article reads as a thought experiment and remains too theoretical to recommend publication in the current state. I am detailing below some of my major comments to improve this manuscript.

Response/action: We appreciate your valuable opinions regarding writing and organization of the manuscript, and have adopted these suggestions in the revised manuscript. We clarify that the present model is designed specifically to study the impact of MICP on LDMP settling. Previous studies on the MICP on LDMPs are referenced, and we have combined the methodology in the method section and Text S1 to form a complete method section with more details provided. Furthermore, additional references have been added to justify MICP and LDMPs.

Major comments:

In the introduction, a paragraph is missing to explain why calcite precipitation is important for plastic debris distribution. References are needed here to justify that this phenomenon is worth investigating and exploring in a model. This would justify the present study.

Many sentences lack references, as a result, a lot of the arguments presented remain pure speculation. For instance, lines 89-94 “The distribution of plastic debris within the second

accumulation zone is a dynamic interplay, intricately influenced by multifarious variables such as incident light intensity, nutrient level, prevailing seawater condition, distance from emission sources, and the distinctive attributes of said sources. Consequently, one may observe a spectrum of concentrations ranging from several to hundreds of plastic particles per cubic meter.” But also lines 109-115, 190-200.

Response/action: The reviewer's concerns are well taken. In the revised manuscript, a new paragraph regarding the calcite precipitation is provided. Additionally, references and extra statements have been added as suggested, focusing on the impact of microbially induced calcium carbonation precipitation on the vertical settling of low-density microplastics and removing speculations on horizontal effects. We also adjust the order of discussions to improve the logic flows, which makes it easier to understand than the previous version. Finally, improper statements and arguments without supporting information (observation or model simulation) have been removed (e.g., lines 190–200).

"Microbially induced calcium carbonate precipitation (MICP) is commonly observed under calcium-rich and high pH environments, such as ocean surface and soil matrix³²⁻³⁴. The present study considers MICP accompanied by biofouling (autotrophic path)³⁵. Most algae have negative surface charges³⁶, and continue attracting over-saturated Ca^{2+} in the upper seawater column^{36,37}. Some biological processes, such as photosynthesis and hydrolysis of urea, can increase pH levels in surrounding water³³, triggering CaCO_3 to precipitate near the biofilm and become part of the biofilm exoskeleton^{32,38}. Different types of CaCO_3 , including calcite and aragonite³⁹, can be produced by MICP. However, aragonite formation requires temperatures over 30 °C⁴⁰. Thus, calcite is considered the primary precipitate on LDMPs⁴¹. With a density over 2.63 g cm⁻³, calcite could be an adequate ballast to sink LDMPs to ocean sediments." (lines 71–80)

"The first accumulation zone occurs at the upper water column near the sea surface (60–400 m), coinciding with assumed static seawater conditions. Fischer et al.²⁵ and Lobelle et al.³¹ demonstrated that incorporating oceanographic processes such as large-scale 3D advection, small-scale vertical turbulence, dynamic grazing, and wind-induced mixing could spread LDMPs to much deeper waters (up to 5,500 m). Still, the maximum concentration of LDMPs remains in the upper water column⁶⁸. It seems that the oceanographic processes can assist in creating dynamic and complex scenarios for LDMPs." (lines 241–247)

"The subsurface accumulation zone commonly appears in subtropical oceans^{17,18}, where the chlorophyll concentration and light intensity are consistent throughout the year (Figs. 5A–5B), creating relatively stable conditions for biofouling and MICP^{69,70}. As seawater density increases with increasing depth, it is difficult for large-sized (> 500 μm) spherical LDMPs to settle as they cannot accumulate enough ballast and are sensitive to density gradients (Fig. 4D). Most of these LDMPs stop settling and concentrate at the subsurface zone⁶⁸." (lines 248–253)

"As the settling time passes the 100th day, most LDMPs begin to diffuse into the deep water column, and the subsurface accumulation zone begins to fade away due to lack of continuous supply of LDMPs (Fig. 6B). Spheres occur less commonly in the seafloor than fibers and films. Starting from the 200th day, most LDMPs have already sunk to the seafloor

(Fig. 6C). On the 300th day, the distribution of LDMPs shows a gradually dispersed trend, with few LDMPs remaining in the water column (Fig. 6D)." (lines 304–309)

One thing I found particularly confusing is that the dimensions of the plastic debris of interest in this study are not described until line 118. As a result, we do not know what kind of plastic debris are simulated and measured.

Response/action: The reviewer's concerns are well taken. In the revised manuscript, an additional statement regarding the dimensions of the plastic debris is provided in the Introduction section, which may reduce the confusion.

"Plastic debris conform to a fractal process through aging and fragmentation upon photolysis and abrasion, breaking into smaller pieces⁵. However, field measurements confirmed that the estimated amount of low-density plastic debris discharged into the global ocean was much greater than that observed on the oceanic surface, especially for plastic debris with size below 5 mm, also known as low-density microplastics (LDMPs)^{6,7}." (lines 33–38)

Lines 75-80: what data are you referring to? You also state that "The fitting performance indicates that improvements are achieved in simulating the vertical settling dynamics of plastic debris in open oceans." compared to what? How can you affirm that the vertical dynamics are improved?

Response/action: The reviewer's concerns are well taken. We totally agree with the reviewer that the comparison here is improper, and the statement is confusing. The Fréchet similarity may not be convincing since the present model is merely an 1D vertical model that only considers biofouling and calcite precipitation. The simulated distribution of plastic debris would deviate from reality without consideration of other oceanographic processes. In the revised manuscript, we use the calcite precipitation theory to explain the gathering of LDMPs in the upper water column and oceanic sediment. Such gatherings have been confirmed via field investigations in recent years.

"Considerable amounts of LDMPs were detected from the twilight zone to the abyssal seafloor^{15,16}. In seawater, LDMPs were detected at the 1,000-m depth, peaking at 200–400 m at an off-shore site close to California, USA¹⁷. Li et al.¹⁸ reported LDMPs at a spate of vertical depths up to 2,000 m at six locations in the Pacific Ocean and the Indian Ocean. Nakajima et al.¹⁹ also reported the widespread occurrence of benthic plastic debris in the deep-sea basin of the Northwest Pacific (3,500–6,500 m), up to 70% of which are LDMPs." (lines 43–48)

"The vertical distribution of LDMPs in the Arctic Ocean is unique compared to those in other oceans, and the present model can well explain this unique distribution of LDMPs. The concentration of LDMPs in the Arctic Ocean is higher at both the sea surface and oceanic sediment (seafloor) and lower in the upper water column (Fig. 5A)^{1,13,65}. No subsurface accumulation zone is observed, which could be attributed to the dramatic seasonal variations in light intensity and chlorophyll concentration⁷³. In summer, light intensity and chlorophyll

concentration at the Arctic Ocean reach their maximums, and biofouling and MICP quickly uplift the density of LDMPs (Fig. 5B). Once overcoming the vertical drag coefficient, LDMPs can settle to the seafloor. The seawater density of the Arctic Ocean is almost consistent from the surface to the bottom (Fig. 5C), posing the least hindering effect on the settling of LDMPs⁷⁴.

There are also vertical downward currents in the Arctic Ocean⁷⁵, which further promote the settling of LDMPs, especially those with a size below 10 μm . In winter, however, light intensity is almost zero. LDMPs in the subsurface zone would ascend due to the lack of biofouling and MICP, which explains the high abundances of LDMPs on the ocean surface¹ (Figs. 5D–5E). Seasonal alternation creates a relatively long period of time with no light, which means that the aggregation of LDMPs is insufficient due to inadequate ballast. As a result, the most favorable size range for vertical settling of LDMPs should be 10–500 μm based on the present model, which is consistent with the results from sediment core analyses^{12,13} (Fig 5F) that the most abundant LDMPs were in the 11–200 μm size class⁶⁵. The aging and fragmentation may also explain the absence of large LDMPs during long-range transport through the global ocean circulation^{61,67,76,77}." (lines 267–287)

In general, the figures could be improved. Many lack some units or information to help interpret their results. Moreover, the text describing the figures presents things that we can't see on them. For instance lines 103-104: "Once the density of plastic debris surpasses the seawater density and overcome the seawater viscosity, plastic debris can settle on the seafloor (Fig. 1)" There is no mention of seawater density or viscosity in the figure.

Response/action: The reviewer's concerns are well taken. In the revised manuscript, the figure has been improved with a clear comparison of seawater density (Fig. 4). The viscosity is represented as the drag coefficients, which will also be presented in the new figure.

"With growing ballast (biofilm and calcite), LDMP density surpasses the surrounding seawater density, triggering initial settling of LDMPs to the water column." (lines 111–113)

Lines 177-179 (describing Fig.4): "This trend reveals that the settling of plastic debris is increasingly dependent on calcite precipitation and biofilm growth as the equivalent diameter of plastic debris becomes smaller due to photolysis and abrasion." No mention of photolysis and abrasion anywhere in the figure (or the MS).

Response/action: The reviewer's concerns are well taken. In the revised manuscript, the impact of photolysis and abrasion has connected to the fractal process with reference. The figure instruction only includes the simulated results.

"Plastic debris conform to a fractal process through aging and fragmentation upon photolysis and abrasion, breaking into smaller pieces⁵. However, field measurements confirmed that the estimated amount of low-density plastic debris discharged into the global ocean was much greater than that observed on the oceanic surface, especially for plastic debris with size below 5 mm, also known as low-density microplastics (LDMPs)^{6,7}." (lines 33–38)

Figure 1: What's "ss" on the plots? How were the curves calculated? They seem far from the dots at some sites (e.g. North Pacific Ocean). No units on the x-axis.

Response/action: The reviewer's concerns are well taken. The 'ss' represents the similarity between simulation and observation. However, since no direct similarity comparison is presented in the revised manuscript, the curves fitting and 'ss' are removed. The figure is also modified to exhibit more observations. It is now presented as Fig. 5.

Figure 4. What's the shape category of the particles in A, B, C, G, H and I. How does this choice affect the results?

Response/action: The reviewer's concerns are well taken. The shape categories of the LDMPs in A, B, C, G, H, and I are evenly assigned as sphere, film, and fiber. It is now presented as Figure 3.

"The shape categories of LDMPs (Figs. 3A–3I) are evenly assigned as sphere, film, and fiber." (lines 174–175)

Figure 5 must be modified. As presented, we cannot distinguish the 3 shapes. What's the density of the particles? What's the simulation conditions?

Response/action: The reviewer's concerns are well taken. The simulation conditions and density of LDMPs are given as follows. It is now presented as Fig. 6.

"Figure 6 shows the settling distribution of a group of typical LDMPs in a time series with a size range of 1–5000 μm , density range of 0.85–1.00 g cm^{-3} , and shape categories of sphere, film, and fiber. The simulation conditions are settled as the tropical Pacific Ocean profiles." (lines 290–292)

Lines 184-189: "Moreover, sphere-shaped plastic debris are most likely to accumulate on the seafloor. Hence fibers and films are more abundant than spheres in the water column. A close examination of plastic debris sized below 1 mm indicates that fibers and films need much longer time than spheres to reach the seafloor, especially for those with equivalent diameters smaller than 100 μm , even though they take a shorter time to submerge into the water column." While this may be shown in your simulations, such results are not new and where seen by other authors. References and discussion of your results with regards to these previous studies is lacking here.

Response/action: The reviewer's concerns are well taken. In the revised manuscript, other studies regarding the abundance of film and fibers in the water column are referenced.

"A close examination of simulated LDMPs sized below 1 mm indicates that fibers and films need much longer time than spheres to reach the seafloor, especially for those with equivalent diameters smaller than 100 μm . This trend has been observed in several previous

studies, and the present model may provide a viable explanation for the high abundances of fibers and films in the water column⁵⁸⁻⁶⁰." (lines 193–197)

Line 253: "In summary, large plastic debris (> 200 μm) are concentrated at 60–400 m and on the seafloor" This conclusion (which is a major result of the study since it's also highlighted in the abstract) seems supported by none of the figures presented. Maybe in Figure 4 which presents results from theoretical simulations.

Response/action: The reviewer's concerns are well taken. In the revised manuscript, a new figure regarding the size and settling is provided, and the discussion is improved due to the consideration of other oceanographic processes.

"Spheres are more concentrated at 100–200 m, where fibers and films distribute in a much deeper and broader region of 100–400 m (Figs. 3D–3F). The wider distribution of fibers and films compared to spheres can be explained by the uncertainty of the shape factor due to the extra degree of freedom of motion. Unlike spheres, fibers, and films can twist and flip naturally during settling, so the projection area towards the settling direction could vary accordingly, which creates deviations in the vertical drag coefficient⁵⁷." (lines 185–191)

"Sphere-shaped LDMPs in a size range of 100–500 μm can directly settle to the ocean sediment, and their settling velocities (100–300 m day^{-1}) are much higher than marine snow (68 m day^{-1})⁶² and fecal pellet (25–67 m day^{-1})⁶³. For sphere-shaped LDMPs in the size range of 10–100 μm , the settling velocity is reduced to 10–20 m day^{-1} . Increased vertical drag coefficient for LDMPs in the size range below 10 μm would further decrease the settling velocity to less than 1 m day^{-1} (Fig. 4D). The high abundance of LDMPs with the size range below 10 μm is inconsistent with field measurements. Moreover, LDMPs in the size range of 1–5 mm cannot settle to the seafloor, which is consistent with the observed size distribution of LDMPs at the ocean sediment.

In previous studies, aggregation has been demonstrated as one of the important processes assisting the settling of ocean particles⁵⁰, which may also boost the settling of LDMPs with the size ranges of <10 μm and 1–5 mm^{35,64}. The simulation results indicate that such aggregation could sink a significant amount of LDMPs in the size range below 10 μm , aggregating them with portions of LDMPs in the size range of 1–5 mm. On one hand, oscillating large-sized LDMPs have more chances to collide with other LDMPs. On the other hand, small-sized LDMPs are much more abundant according to the exponential size-distribution of LDMPs. With high-density fouling, these small-sized LDMPs are the perfect ballast to aggregate with large-sized LDMPs. Aggregation of LDMPs with marine snow and fecal pellets also occurs, and the aggregates could settle to the seafloor under a small vertical drag coefficient." (lines 211–229)

The sentence line 258-261 "However, such depletion of plastic debris in the upper 100 m water column can hardly be field-measured, because increasing plastic production and insufficient plastic waste management constantly supplement plastic debris to the sea surface." is awkward. It seems to justify that the results of this study may not be supported by data.

Response/action: The reviewer's concerns are well taken. In the revised manuscript, this statement is removed.

Line 265: "Since it requires a considerable amount of time for plastic debris to build up calcite precipitates and biofilms" This sentence lacks a reference, and is not true. Biofilms form within minutes to hours (see Paul-Pont et al., 2018)

Paul-Pont I, Tallec K, Gonzalez-Fernandez C, Lambert C, Vincent D, Mazurais D, Zambonino-Infante J-L, Brotons G, Lagarde F, Fabioux C, Soudant P and Huvet A (2018) Constraints and Priorities for Conducting Experimental Exposures of Marine Organisms to Microplastics. *Front. Mar. Sci.* 5:252. doi: 10.3389/fmars.2018.00252

Response/action: The reviewer's concerns are well taken. In the revised manuscript, the statement has been corrected as suggested.

"Depending on the algal concentration and kernel encounter rate^{3,50}, biofilm forms within minutes to hours⁵¹." (lines 110–111)

"The accumulation of biofilm and MICP takes hours or days to complete, which is consistent with field observations⁵⁴." (lines 177–178)

Lines 267-273: "By analyzing the vertical distribution patterns of plastic debris, we can diagnose possible emission sources and their geographic locations (Fig. S3). If plastic debris collected at a given geographic location originate from nearby emission sources, the surface concentration of plastic debris would be higher than that resulting from distant emission sources. If plastic debris are derived from multiple emission sources with numerous distances, their vertical distribution patterns would contain multiple overlapping peaks. Again, no reference for these affirmations. Moreover, these affirmations completely ignore the fact that ocean currents may also be vertical! Vertical currents have been shown to significantly influence plastic distribution in 3D (Richon et al., 2022, Mountford et al., 2019, Huck et al., 2022). Also, these sentences overlook the potential influence of fish and other animals on plastic vertical distribution through ingestion-migration-excretion (Justino et al., 2020).

Justino, A. K. S., Ferreira, G. V. B., Schmidt, N., Eduardo, L. N., Fauvelle, V., Lenoble, V., Sempéré, R., Panagiotopoulos, C., Mincarone, M. M., Frédou, T., & Lucena-Frédou, F. (2022). The role of mesopelagic fishes as microplastics vectors across the deep-sea layers from the Southwestern Tropical Atlantic. *Environmental pollution (Barking, Essex : 1987)*, 300, 118988. <https://doi.org/10.1016/j.envpol.2022.118988>

Richon C, Gorgues T, Paul-Pont I and Maes C (2022) Zooplankton exposure to microplastics at global scale: Influence of vertical distribution and seasonality. *Front. Mar. Sci.* 9:947309. doi: 10.3389/fmars.2022.947309

Huck T, Bajon R, Grima N, Portela E, Molines J-M and Penduff T (2022) Three-Dimensional Dispersion of Neutral “Plastic” Particles in a Global Ocean Model. *Front. Anal. Sci.* 2:868515. doi: 10.3389/frans.2022.868515

Mountford, A. S., & Morales Maqueda, M. A. (2019). Eulerian Modeling of the Three-Dimensional Distribution of Seven Popular Microplastic Types in the Global Ocean. *Journal of Geophysical Research: Oceans*, 124, 8558–8573. <https://doi.org/10.1029/2019JC015050>

Response/action: The reviewer's concerns are well taken. We realize that discussing the vertical distribution of LDMPs without considering other oceanographic processes is problematic. In the revised manuscript, this paragraph is deleted. Instead, several improvements have been made to strengthen the discussion. Moreover, the calcite precipitation is not an exclusive theory, it can combine with other oceanographic processes to reinforce the settling of LDMPs. In fact, we have added extra discussion to describe these co-effects.

"However, the biofouling model encounters enormous challenges to explain the enrichment of LDMPs in the oceanic sediment. Modeled LDMPs were impossible to approach the ocean floor due to the continuing loss of biofilm²⁶. Fischer et al.²⁵ evaluated LDMP settling under another biofilm dynamic, assuming frustule attachment on LDMPs after settling breathe the twilight zone. Ingestion by marine organisms (fecal pellets) merely sinks a negligible fraction (0.13–0.19%) of LDMPs^{27,28}. Aggregation with biogenic particles (marine snow) could sink another 0.06–8.8% of LDMPs^{29,30}. Lobelle et al.³¹ and Fischer et al.²⁵ incorporated oceanographic processes with a settling model, including large-scale 3D advection, small-scale vertical turbulence, dynamic grazing, and wind-induced mixing, etc. The updated model allowed LDMPs to sink below the euphotic zone and mixed layer; however, only 15 of 10,000 LDMPs reached the ocean floor (>5,000 m) under favorable conditions²⁵." (lines 60–70)

"In previous studies, aggregation has been demonstrated as one of the important processes assisting the settling of ocean particles⁵⁰, which may also boost the settling of LDMPs with the size ranges of <10 μm and 1–5 mm^{35,64}. The simulation results indicate that such aggregation could sink a significant amount of LDMPs in the size range below 10 μm, aggregating them with portions of LDMPs in the size range of 1–5 mm. On one hand, oscillating large-sized LDMPs have more chances to collide with other LDMPs. On the other hand, small-sized LDMPs are much more abundant according to the exponential size-distribution of LDMPs. With high-density fouling, these small-sized LDMPs are the perfect ballast to aggregate with large-sized LDMPs. Aggregation of LDMPs with marine snow and fecal pellets also occurs, and the aggregates could settle to the seafloor under a small vertical drag coefficient." (lines 220–229)

"The first accumulation zone occurs at the upper water column near the sea surface (60–400 m), coinciding with assumed static seawater conditions. Fischer et al.²⁵ and Lobelle et al.³¹ demonstrated that incorporating oceanographic processes such as large-scale 3D advection, small-scale vertical turbulence, dynamic grazing, and wind-induced mixing could

spread LDMPs to much deeper waters (up to 5,500 m). Still, the maximum concentration of LDMPs remains in the upper water column⁶⁸. It seems that the oceanographic processes can assist in creating dynamic and complex scenarios for LDMPs." (lines 241–247)

"The vertical distribution of LDMPs in the Arctic Ocean is unique compared to those in other oceans, and the present model can well explain this unique distribution of LDMPs. The concentration of LDMPs in the Arctic Ocean is higher at both the sea surface and oceanic sediment (seafloor) and lower in the upper water column (Fig. 5A)^{1,13,65}. No subsurface accumulation zone is observed, which could be attributed to the dramatic seasonal variations in light intensity and chlorophyll concentration⁷³. In summer, light intensity and chlorophyll concentration at the Arctic Ocean reach their maximums, and biofouling and MICP quickly uplift the density of LDMPs (Fig. 5B). Once overcoming the vertical drag coefficient, LDMPs can settle to the seafloor. The seawater density of the Arctic Ocean is almost consistent from the surface to the bottom (Fig. 5C), posing the least hindering effect on the settling of LDMPs⁷⁴.

There are also vertical downward currents in the Arctic Ocean⁷⁵, which further promote the settling of LDMPs, especially those with a size below 10 μm . In winter, however, light intensity is almost zero. LDMPs in the subsurface zone would ascend due to the lack of biofouling and MICP, which explains the high abundances of LDMPs on the ocean surface¹ (Figs. 5D–5E). Seasonal alternation creates a relatively long period of time with no light, which means that the aggregation of LDMPs is insufficient due to inadequate ballast. As a result, the most favorable size range for vertical settling of LDMPs should be 10–500 μm based on the present model, which is consistent with the results from sediment core analyses^{12,13} (Fig 5F) that the most abundant LDMPs were in the 11–200 μm size class⁶⁵. The aging and fragmentation may also explain the absence of large LDMPs during long-range transport through the global ocean circulation^{61,67,76,77}." (lines 267–287)

One major question that I have is if biofilm detaches when plastic sinks, why would calcite precipitates remain on plastic surface? This is not justified anywhere in the manuscript.

Response/action: The reviewer's concerns are well taken. Studies on calcite precipitation on LDMPs are referenced in the revised manuscript.

"Moreover, calcite precipitates are difficult to come off plastic surfaces³⁴. Laboratory studies revealed that the binding between calcite and LDMPs is so strong that calcite can be used as a coat to strengthen plastics in cementitious materials^{37,50,51}." (lines 100–103)

Methods

In general, the methods (except for the model description and equations) lack important information. There is no information on whether the model used in this study was built specifically, or if it existed before. Again, there is no reference to calcite precipitation being a significant process (lines 309-311 have no reference).

Response/action: The reviewer's concerns are well taken. The present model is designed specifically to study the impact of MICP on LDMP settling. The missing information in the methods section is presented in the supplemental material. In the revised manuscript, studies on the MICP on LDMPs are referenced, and we have combined the methodology in the method section and Text S1 to form a complete method description.

"Moreover, calcite precipitates are difficult to come off plastic surfaces³⁴. Laboratory studies revealed that the binding between calcite and LDMPs is so strong that calcite can be used as a coat to strengthen plastics in cementitious materials^{37,50,51}." (lines 100–103)

Line 364-365: "Calcite precipitates are much denser than seawater and are difficult to detach, thus they are much more efficient than algae in increasing the density of plastic debris." Needs a reference. Is there any study showing calcite precipitation on plastic debris?

Response/action: The reviewer's concerns are well taken. Studies on the calcite precipitation on plastic debris are referenced in the revised manuscript. There are several studies discussing the MICP on LDMPs, which have been added as references.

"Microbially induced calcium carbonate precipitation (MICP) is commonly observed under calcium-rich and high pH environments, such as ocean surface and soil matrix³²⁻³⁴. The present study considers MICP accompanied by biofouling (autotrophic path)³⁵. Most algae have negative surface charges³⁶, and continue attracting over-saturated Ca^{2+} in the upper seawater column^{36,37}. Some biological processes, such as photosynthesis and hydrolysis of urea, can increase pH levels in surrounding water³³, triggering CaCO_3 to precipitate near the biofilm and become part of the biofilm exoskeleton^{32,38}. Different types of CaCO_3 , including calcite and aragonite³⁹, can be produced by MICP. However, aragonite formation requires temperatures over 30 °C⁴⁰. Thus, calcite is considered the primary precipitate on LDMPs⁴¹. With a density over 2.63 g cm⁻³, calcite could be an adequate ballast to sink LDMPs to ocean sediments." (lines 71–80)

"Moreover, calcite precipitates are difficult to come off plastic surfaces³⁴. Laboratory studies revealed that the binding between calcite and LDMPs is so strong that calcite can be used as a coat to strengthen plastics in cementitious materials^{37,50,51}." (lines 100–103)

Lines 387-406: what location did you use? How did you select the sampling sites?

Response/action: The reviewer's concerns are well taken. We apologize for the incomplete statement regarding the simulation location used in the manuscript. In the revised manuscript, all the comparison sampling sites are listed in Table S1.

"The Monte Carlo simulation methodology was introduced, and a group of simulated LDMPs was used to study the distribution patterns of LDMPs along the vertical direction based on available field investigations (Table S1). Because the exact number of LDMPs imported to the specific geographic location is almost impossible to acquire, the comparison mainly focused on the characteristics of LDMPs vertical settling patterns rather than the exact concentration distribution of LDMPs. To maximize the proximity to reality, the vertical seawater salinity and temperature profiles are acquired with latitude and longitude as the

sampling location given by the available field measurements^{17,18,65,66} through the global real-time ocean forecasting system (RTOFS)^{70,92,74}." (lines 647–655)

Line 395-397: "We may move the coordinate of simulated seawater profiles a little bit from the original position if the depth of the RTOFS dataset at the original site cannot approach the measured depth" Since we have no idea of the locations you are referring to, this sentence makes no sense.

Response/action: The reviewer's concerns are well taken. In the revised manuscript, this sentence is removed since we focus on the general vertical distributions in the observations rather than fitting these concentrations.

The information of size and density of the plastics should also be mentioned in the main manuscript.

Response/action: The reviewer's concerns are well taken. In the revised manuscript, information on the size and density of the plastics is added.

"In the present study, a new one-dimensional hydrodynamic model is designed to evaluate the impact of MICP on LDMP settling. The MICP is controlled by the algal photosynthetic intensity. The target LDMPs have a density of 0.85–1.00 g cm⁻³ with an equivalent size range between 1 μm and 5 mm (see Methods), following an exponential distribution⁴². Sphere, fiber (57–13,000 μm in length), and film (0.1–100 μm in thickness) are evenly distributed." (lines 81–85)

Lines 405-406: "The floating plastic debris which are denser than the seawater may be caused by trapped air, hydrophobicity, and surface tension." So, you simulated bubble formation?

Response/action: The reviewer's concerns are well taken. In the previous version, this description was used to assist the explanation of slightly heavy plastics on the surface. Then we realized that the primary target is low-density microplastics. In the revised manuscript, this sentence is removed since no supporting study is presented in the manuscript.

Lines 409-420: Again, no mention of the observations, what observations? Where? Who sampled? How was it sampled? I don't understand the organization of the model with colored modules.

Response/action: The reviewer's concerns are well taken. The observation of vertical distribution of LDMPs is in the beginning of the result and discussion section and the supplemental materials. We apologize for the incomplete statement regarding the simulation location used in the manuscript. In the revised manuscript, all the comparison sampling sites are listed in Fig. 5 and Table S1.

"The Monte Carlo simulation methodology was introduced, and a group of simulated LDMPs was used to study the distribution patterns of LDMPs along the vertical direction

based on available field investigations (Table S1). Because the exact number of LDMPs imported to the specific geographic location is almost impossible to acquire, the comparison mainly focused on the characteristics of LDMPs vertical settling patterns rather than the exact concentration distribution of LDMPs. To maximize the proximity to reality, the vertical seawater salinity and temperature profiles are acquired with latitude and longitude as the sampling location given by the available field measurements^{17,18,65,66} through the global real-time ocean forecasting system (RTOFS)^{70,92,74}." (lines 647–655)

Lines 421-422: "The vertical distribution patterns of plastic debris are estimated via the interpolation method to create smooth curves for comparison." Comparison with what? What interpolation method?

Response/action: The reviewer's concerns are well taken. We totally agree with the reviewer that the comparison here is improper, and the statement is confusing. The Fréchet similarity may not be convincing since the present model is merely an 1D vertical model that only considers biofouling and calcite precipitation. The simulated distribution of LDMPs would deviate from reality without considering other oceanographic processes. In the revised manuscript, we delete the contents regarding the directed comparison between curved observation and simulation using the Fréchet similarity. Instead, we use the calcite precipitation theory to explain the gathering of LDMPs in upper water column and ocean sediment with detailed discussion. Such gatherings have been confirmed via field investigations in recent years.

"Considerable amounts of LDMPs were detected from the twilight zone to the abyssal seafloor^{15,16}. In seawater, LDMPs were detected at the 1,000-m depth, peaking at 200–400 m at an off-shore site close to California, USA¹⁷. Li et al.¹⁸ reported LDMPs at a spate of vertical depths up to 2,000 m at six locations in the Pacific Ocean and the Indian Ocean. Nakajima et al.¹⁹ also reported the widespread occurrence of benthic plastic debris in the deep-sea basin of the Northwest Pacific (3,500–6,500 m), up to 70% of which are LDMPs." (lines 43–48)

"The vertical distribution of LDMPs in the Arctic Ocean is unique compared to those in other oceans, and the present model can well explain this unique distribution of LDMPs. The concentration of LDMPs in the Arctic Ocean is higher at both the sea surface and oceanic sediment (seafloor) and lower in the upper water column (Fig. 5A)^{1,13,65}. No subsurface accumulation zone is observed, which could be attributed to the dramatic seasonal variations in light intensity and chlorophyll concentration⁷³. In summer, light intensity and chlorophyll concentration at the Arctic Ocean reach their maximums, and biofouling and MICP quickly uplift the density of LDMPs (Fig. 5B). Once overcoming the vertical drag coefficient, LDMPs can settle to the seafloor. The seawater density of the Arctic Ocean is almost consistent from the surface to the bottom (Fig. 5C), posing the least hindering effect on the settling of LDMPs⁷⁴.

There are also vertical downward currents in the Arctic Ocean⁷⁵, which further promote the settling of LDMPs, especially those with a size below 10 μm . In winter, however, light intensity is almost zero. LDMPs in the subsurface zone would ascend due to the lack of

biofouling and MICP, which explains the high abundances of LDMPs on the ocean surface¹ (Figs. 5D–5E). Seasonal alternation creates a relatively long period of time with no light, which means that the aggregation of LDMPs is insufficient due to inadequate ballast. As a result, the most favorable size range for vertical settling of LDMPs should be 10–500 μm based on the present model, which is consistent with the results from sediment core analyses^{12,13} (Fig 5F) that the most abundant LDMPs were in the 11–200 μm size class⁶⁵. The aging and fragmentation may also explain the absence of large LDMPs during long-range transport through the global ocean circulation^{61,67,76,77}." (lines 267–287)

The most problematic with the methods section is that there is no mention of how the simulations were performed. How long did they run for? Is that 0D, 1D, 2D, 3D? What data is used for forcing? Is there a control simulation? Did you simulate all processes separately? Did you simulate biofouling & calcite precipitation? Did you perform a simulation without calcite precipitation for comparison? As presented, it seems not. Thus, there is no way to know whether calcite precipitation is the process actually impacting plastic export.

Response/action: The reviewer's concerns are well taken. We agree with the reviewer that the current method section is inadequate, and splitting the methodology into supplemental material breaks the integrity and coherence of the model. In the revised manuscript, we combined both sections and rearranged the Introduction as Method section. Additional descriptions are also added in the Introduction section.

1. The time length for model simulation:

"To fully evaluate the settling patterns of LDMPs, we use the Monte Carlo method to generate 500 random LDMPs, calculate their vertical trajectories with and without MICP using the present model, and record their depths in a time series of 200 days (Fig. 1)." (lines 104–106)

2. The model dimension:

"In the present study, a new one-dimensional hydrodynamic model is designed to evaluate the impact of MICP on LDMP settling." (lines 81–82)

3. Data used for model forcing

"Low-density microplastics under investigation are sized between 1 μm and 5 mm following an exponential distribution, which was adjusted to meet previous observations that the quantity of LDMPs with a size between 100–500 μm is ten times compared to the number of LDMPs generated between 500 μm –5 mm⁸¹. For fiber-like LDMPs, a length range between 100 μm and 1.3 mm is selected. The thickness of film-like LDMPs varies between 0.1–100 μm . Each simulated LDMPs is randomly assigned with an initial density from 850 to 1,000 kg m^{-3} ^{82,83}. A random plastic generator was added to the model, generating LDMPs following the above-mentioned conditions. For simulation purposes, the ratio of plastic debris of different shapes is assumed as fiber: sphere: film = 1:1:1." (lines 350–358)

"The seawater profiles are acquired from the global real-time ocean forecasting system (RTOFS)." (lines 360–361)

"In a typical saturation state of 1.71, the precipitation rate of calcite is estimated as $5 \mu\text{mol cm}^{-2} \text{s}^{-1}$ ³³." (lines 427–428)

"To study the impact of CSD, we introduce the calcite dissolution process when the adopted seawater condition shows the sinking depth is below the CSD³²." (lines 443–445)

"In the present study, only the nowcasting dataset is adopted. We have collected the nowcasting dataset from December 18, 2021, to December 18, 2022 using an auto-web crawler programmed by Python 3.7. Both salinity and temperature profiles include 33 available depths from the sea surface to the seafloor." (lines 612–615)

4. Is there a control simulation?

Yes. The control simulation of the model is described in the revised manuscript.

"The present model calculates the vertical settling speed and trajectory of LDMPs, considering the initial density, shape, and size of LDMPs and designated ocean conditions. The present model is a one-dimensional depth-dependent calcium carbonate (CaCO_3) precipitation model using dynamic seawater conditions in a quiescent ocean based on Kooi's model, which is explicitly designed to examine the impact of MICP during plastic settling." (lines 344–349)

5. Did you simulate all processes separately?

Yes. The process included in the model is described in the revised manuscript.

"The present model includes three complementary sub-models: An updated vertical settling model, a MICP model, and a biofouling model. The vertical settling model based on fluid dynamic principles describes the settling speed and motion of unregulated LDMPs under a wide range of Reynolds numbers, taking into account the temporary density of LDMPs and ocean profiles. The MICP model was designed to estimate calcite precipitates on the plastic surface, considering the activity level of algae photosynthesis and dissolution rate under CSD. The biofouling model simulates biofilm growth on LDMPs, established by Kooi et al.²², which includes algae attachment, growth, mortality, and respiration under specific nutrition conditions and light intensity. Key parameters are listed in the supplementary materials (Table S2)" (lines 362–367)

6. Did you simulate biofouling & calcite precipitation?

Yes. The biofouling and calcite precipitation modules are described in the revised manuscript.

"3. The MICP model

Microbially induced calcium carbonate precipitation relies on biological processes, which are highly dependent on the presence of bacteria³⁷. Kooi et al.²² pointed out that the number and weight of bacteria are much less than algae. Thus, the present study only considers autotrophic MICP and calcite as modeling processes. The calcium flux (J ; $\mu\text{mol cm}^{-2} \text{s}^{-1}$) at the plastic surface is estimated using Fick's first law of diffusion³⁸:..." (lines 418–422)

"4. The biofouling growth model

4.1. Biofouling formation in open oceans

The biofouling formation on the material surface is a combination of microorganisms, including bacteria, algae, and/or fungi⁸⁷. For LDMPs, the biota types on the biofouling cover may be related to the original size, shape, and surface roughness of LDMPs. The biofouling process, according to previous studies, can be described as follows^{88,89}..." (lines 449-452)

7. Did you perform a simulation without calcite precipitation for comparison?

Yes. The comparison with and without calcite precipitation is given in the manuscript.

"The settling patterns of LDMPs in the deep ocean can be substantially altered by MICP, as compared to the results by Kooi's model, which only considered biofouling process (Fig. 1A). Shortly after being released to simulated seawater, algae begin to collide and attach on the LDMP surface. Depending on the algal concentration and kernel encounter rate^{3,50}, biofilm forms within minutes to hours⁵¹. During the process, MICP delivers calcite on the plastic surface. With growing ballast (biofilm and calcite), LDMP density surpasses the surrounding seawater density, triggering initial settling of LDMPs to the water column. Due to light intensity attenuation with increasing depth, biofilm decays through dying, shrinking, and shedding⁵². Although MICP also fades due to reduced pH caused by weakened algal photosynthesis^{32,53}, calcite precipitates are more difficult to detach from the plastic surface than biofilms. Thus, LDMPs can retain most of the gained density and continue to settle until the negative buoyancy is balanced with the vertical drag. Depending on the settling conditions, most LDMPs could sink to the seafloor within tens to hundreds of days (Fig. 1B). Microbially induced calcium carbonate precipitation may serve as one of the critical factors in depleting LDMP concentrations at the ocean subsurface and enhancing LDMP accumulation at the oceanic sediment¹⁶." (lines 106–121)

Minor corrections:

- Line 50: "Indian Ocean"

Response/action: The reviewer's concerns are well taken. In the revised manuscript, the statement has been corrected as suggested.

"Li et al.¹⁸ detected LDMPs at a spate of vertical depths up to 2,000 m at six locations in the Pacific Ocean and the Indian Ocean." (lines 45–46)

- Line 146: What's the "plastic debris simulator"?

Response/action: The reviewer's concerns are well taken. The plastic debris simulator is a module in the present model that can generate plastic debris, meeting the initial density, shape, and size required for model simulation. In the revised manuscript, the description is moved to the Method section to avoid further confusion.

"A random plastic generator was added to the model, generating LDMPs following the above-mentioned conditions. For simulation purposes, the ratio of plastic debris of different shapes is assumed as fiber: sphere: film = 1:1:1." (lines 356–358)

- Lines 191-193: "Because spheres have the lowest specific surface areas among all shape categories, calcite precipitates, and biofilms are much thicker on spheres than on fibers and films when similar overall density is achieved." Needs a reference

Response/action: The reviewer's concerns are well taken. In the revised manuscript, a reference regarding the specific surface area is added.

"For relatively large LDMPs with a shape factor close to the sphere (the smallest specific surface area under the same volume)⁵⁶, the damped oscillation pattern is frequently observed in the model simulation results (Figs. 2A–2B)." (lines 142–144)

Reviewer 3:

Review of “Sinking the unsinkable: Ballast effects of calcite on low-density plastic debris in open oceans”

In this study, the authors develop a new model of explicit calcite biofouling of microplastics to investigate the transport potential of calcite ballast for moving plastic particles into the deeper ocean and away from the sea surface. They find that the combination of organic and calcite biofouling transports microplastics into the deeper ocean levels and leads to the accumulation of debris in the mesopelagic and near the seafloor. Deeper transport of microplastics is more consistent with observations. Their implementation of multiple plastic particle characteristics in addition to organic and calcite biofouling makes this a novel study that could be interesting to a broad audience. However, some of the arguments put forth are not fully convincing and more information is needed. Furthermore, the organization of the manuscript could be improved for readability. Starting the results section with the Monte Carlo simulations (from L141). Move Fig. 3 to Fig. 1 and remove panel A (or, consider moving Fig. 3 to the supplement and adding non-calcite ballast to current Fig. 2). Make current Fig. 1 new Fig. 2. Add density and chlorophyll profiles to that figure panels. Remove the red curve, add in the no-calcite profile to clearly demonstrate the impact. Move current Fig. 4 up to new Fig. 3. Rearrange text accordingly. I think these changes could improve the flow of the Results section and make it more understandable.

Response/action: The reviewer's suggestions and comments are very constructive. In the revised manuscript, the organization of the manuscript has been greatly improved according to the review's opinions. The result and discussion section begins with the discussion of general impact of MICPs on the settling of LDMPs (moving Fig. 3 to Fig.1 and delete the panel A). Then the manuscript discusses the two primary settling patterns with improved Fig. 2. After that, a sensitivity analysis is performed to discuss the relationship between settling patterns and physical profiles of LDMPs (density, shape, and size, Figs. 3–4). The original Fig.1 is move to Fig 5, which compare the observed accumulation of LDMPs in different ocean areas, explaining the geographic impacts on LDMP vertical distributions. Finally, a Monte Carlo simulation is performed to summarize the general settling of LDMPs in two distinct area.

General comments:

Introduction: The size range of plastic debris should be introduced in the first paragraphs of the manuscript, to make it clear which plastics are being considered. This is also true for the discussion of the Kooi study (from L39), so that the reader understands this is a relevant study to the present topic. Also the paragraph from L44, explicitly stating the size ranges of the observed plastics to make it clear these are fairly compared to the Kooi oscillation mechanism.

Response/action: The reviewer's concerns are well taken. In the revised manuscript, the size range of the LDMPs has been presented in the Introduction section.

"However, repeated field measurements confirmed that the estimated amount of low-density plastic debris discharged into the global ocean was much greater than that observed on the oceanic surface, especially for plastic debris with size below 5 mm, also known as low-density microplastics (LDMPs)^{6,7}." (lines 35–38)

"In the present study, a new one-dimensional hydrodynamic model is designed to evaluate the impact of MICP on LDMP settling. The MICP is controlled by the algal photosynthetic intensity. The target LDMPs have a density of 0.85–1.00 g cm⁻³ with an equivalent size range between 1 µm and 5 mm (see Methods), following an exponential distribution⁴². Sphere, fiber (57–13,000 µm in length), and film (0.1–100 µm in thickness) are evenly distributed." (lines 81–85)

Model validation section:

Figure 1: It would be very helpful to see an annual average chlorophyll map in Fig. 1. This could replace the current map, with the observational points still overlaid. Please remove the red lines from Figure 1 as they do not add any insight into the model assessment. Are the observations down the water column taken at the same time as the surface observations? Please also define the size range of the plastic debris used in this model-obs comparison. And, are the units particles per m⁻³?

The Fréchet distance is not terribly convincing, as the observational curves are not convincingly similar to the observations themselves in many locations. I would prefer a more direct/less derived comparison method.

Response/action: The reviewer's concerns are well taken. In the revised manuscript, an annual average chlorophyll map has been added to replace the previous world map. The red lines are also removed. We have confirmed that the observations down the water column were taken simultaneously with the surface observations in each referenced data. The units of plastic concentration are confirmed as particles per m³ in the water column and particles per kg in the sediment.

We totally agree with the reviewer that the comparison here is improper, and the statement is confusing. The Fréchet similarity may not be convincing since the present model is merely an 1D vertical model that only considers biofouling and calcite precipitation. The simulation distribution of LDMPs would be biased from reality without consideration of other oceanographic processes. Thus, we focus on using the calcite precipitation theory to explain the gathering of LDMPs in upper water column and ocean sediment, such gathering has been confirmed via field investigations in recent years.

"Considerable amounts of LDMPs were detected from the twilight zone to the abyssal seafloor^{15,16}. In seawater, LDMPs were detected at the 1,000-m depth, peaking at 200–400 m at an off-shore site close to California, USA¹⁷. Li et al.¹⁸ reported LDMPs at a spate of vertical depths up to 2,000 m at six locations in the Pacific Ocean and the Indian Ocean.

Nakajima et al.¹⁹ also reported the widespread occurrence of benthic plastic debris in the deep-sea basin of the Northwest Pacific (3,500–6,500 m), up to 70% of which are LDMPs." (lines 43–48)

"The vertical distribution of LDMPs in the Arctic Ocean is unique compared to those in other oceans, and the present model can well explain this unique distribution of LDMPs. The concentration of LDMPs in the Arctic Ocean is higher at both the sea surface and oceanic sediment (seafloor) and lower in the upper water column (Fig. 5A)^{1,13,65}. No subsurface accumulation zone is observed, which could be attributed to the dramatic seasonal variations in light intensity and chlorophyll concentration⁷³. In summer, light intensity and chlorophyll concentration at the Arctic Ocean reach their maximums, and biofouling and MICP quickly uplift the density of LDMPs (Fig. 5B). Once overcoming the vertical drag coefficient, LDMPs can settle to the seafloor. The seawater density of the Arctic Ocean is almost consistent from the surface to the bottom (Fig. 5C), posing the least hindering effect on the settling of LDMPs⁷⁴.

There are also vertical downward currents in the Arctic Ocean⁷⁵, which further promote the settling of LDMPs, especially those with a size below 10 μm. In winter, however, light intensity is almost zero. LDMPs in the subsurface zone would ascend due to the lack of biofouling and MICP, which explains the high abundances of LDMPs on the ocean surface¹ (Figs. 5D–5E). Seasonal alternation creates a relatively long period of time with no light, which means that the aggregation of LDMPs is insufficient due to inadequate ballast. As a result, the most favorable size range for vertical settling of LDMPs should be 10–500 μm based on the present model, which is consistent with the results from sediment core analyses^{12,13} (Fig 5F) that the most abundant LDMPs were in the 11–200 μm size class⁶⁵. The aging and fragmentation may also explain the absence of large LDMPs during long-range transport through the global ocean circulation^{61,67,76,77}." (lines 267–287)

Paragraph starting at L81 is not well justified by the figures. Clarity could be improved by moving Fig. 4 to Fig. 3, which then demonstrates the 2 major accumulation zones for different particle types in context. If chlorophyll is added to Fig. 1 then this would help to demonstrate how biological activity is shaping water column plastic distributions. Why not also add density profiles to the panels of Fig. 1, and then you can remove Fig. S1A and also directly see how the density gradient is affecting accumulation zones? Fig. S1B is not very useful and can be removed.

Response/action: The reviewer's concerns are well taken. In the revised manuscript, the chlorophyll concentration and seawater density differences are added to Fig. 1 (now Fig. 5). Fig. 4 has become Fig. 3 for better clarification. The original Fig. 3 has moved to Fig.1. Additionally, a new Fig. 4 is added to better exhibit the impact of MICP on LDMP settling under various sizes and shapes.

Last 2 sentences from L89 “The distribution of plastic debris...” are vague and not demonstrated clearly from looking at the figures. Fig. 1 appears to show typically lower concentrations.

Response/action: The reviewer's concerns are well taken. In the revised manuscript, the discussion has been improved and the expression has been changed.

"The first accumulation zone occurs at the upper water column near the sea surface (60–400 m), coinciding with assumed static seawater conditions. Fischer et al.²⁵ and Lobelle et al.³¹ demonstrated that incorporating oceanographic processes such as large-scale 3D advection, small-scale vertical turbulence, dynamic grazing, and wind-induced mixing could spread LDMPs to much deeper waters (up to 5,500 m). Still, the maximum concentration of LDMPs remains in the upper water column⁶⁸. It seems that the oceanographic processes can assist in creating dynamic and complex scenarios for LDMPs." (lines 241–247)

"The subsurface accumulation zone commonly appears in subtropical oceans^{17,18}, where the chlorophyll concentration and light intensity are consistent throughout the year (Figs. 5A–5B), creating relatively stable conditions for biofouling and MICP^{69,70}. As seawater density increases with increasing depth, it is difficult for large-sized (> 500 μm) spherical LDMPs to settle as they cannot accumulate enough ballast and are sensitive to density gradients (Fig. 4D). Most of these LDMPs stop settling and concentrate at the subsurface zone⁶⁸. In contrast, the model simulation suggests that LDMPs with a size range below 10 μm can accumulate much heavier ballast (nearly 2.5 g cm^{-3}); however, the high vertical drag coefficient traps these LDMPs in the upper water column for a long time due to extremely slow settling velocity⁷¹. Although aggregation among LDMPs can accelerate the settling process, it is entirely by chance. It is positively correlated with the concentration of LDMPs in the water column, which further assists in forming the subsurface accumulation zone^{35,50,64}." (lines 248–259)

Paragraph starting from L95 is also not completely convincing. Particularly, L98 “No second accumulation zone...”. What about seasonal deep mixing (e.g. Fischer et al. 2022), and the strongly seasonal biomass? Could seasonality be determining the homogeneous distribution? And from the next sentence (L99) Kvale et al. (2020) <https://www.nature.com/articles/s41598-020-72898-4> Supplement has a sensitivity analysis showing that surface concentrations in the Arctic are highly sensitive to the buoyant fraction advecting from lower latitudes, where particles biofouled and in the subsurface below 40N tend not to escape to the Arctic. Can a figure strengthen the arguments put forth in this paragraph?

Response/action: The reviewer's concerns are well taken. In the revised manuscript, brief discussions are provided regarding the seasonal impacts on the vertical distribution of LDMPs in the Arctic Ocean.

"The vertical distribution of LDMPs in the Arctic Ocean is unique compared to those in other oceans, and the present model can well explain this unique distribution of LDMPs. The concentration of LDMPs in the Arctic Ocean is higher at both the sea surface and oceanic sediment (seafloor) and lower in the upper water column (Fig. 5A)^{1,13,65}. No subsurface accumulation zone is observed, which could be attributed to the dramatic seasonal variations in light intensity and chlorophyll concentration⁷³. In summer, light intensity and chlorophyll

concentration at the Arctic Ocean reach their maximums, and biofouling and MICP quickly uplift the density of LDMPs (Fig. 5B). Once overcoming the vertical drag coefficient, LDMPs can settle to the seafloor. The seawater density of the Arctic Ocean is almost consistent from the surface to the bottom (Fig. 5C), posing the least hindering effect on the settling of LDMPs⁷⁴.

There are also vertical downward currents in the Arctic Ocean⁷⁵, which further promote the settling of LDMPs, especially those with a size below 10 μm . In winter, however, light intensity is almost zero. LDMPs in the subsurface zone would ascend due to the lack of biofouling and MICP, which explains the high abundances of LDMPs on the ocean surface¹ (Figs. 5D–5E). Seasonal alternation creates a relatively long period of time with no light, which means that the aggregation of LDMPs is insufficient due to inadequate ballast. As a result, the most favorable size range for vertical settling of LDMPs should be 10–500 μm based on the present model, which is consistent with the results from sediment core analyses^{12,13} (Fig 5F) that the most abundant LDMPs were in the 11–200 μm size class⁶⁵. The aging and fragmentation may also explain the absence of large LDMPs during long-range transport through the global ocean circulation^{61,67,76,77}." (lines 267–287)

Paragraph from L109: Reference to a figure would help here. Better still would be overlaying density and chlorophyll on the panels of Fig. 1.

Response/action: The reviewer's concerns are well taken. In the revised manuscript, a reference is provided in Fig. 4 (original Fig. 1). Moreover, the density and chlorophyll graphs in summer and winter is also provided to better exhibit the site specific conditions regarding the LDMP settling.

"The subsurface accumulation zone commonly appears in subtropical oceans^{17,18}, where the chlorophyll concentration and light intensity are consistent throughout the year (Figs. 5A–5B), creating relatively stable conditions for biofouling and MICP^{69,70}. As seawater density increases with increasing depth, it is difficult for large-sized (> 500 μm) spherical LDMPs to settle as they cannot accumulate enough ballast and are sensitive to density gradients (Fig. 4D). Most of these LDMPs stop settling and concentrate at the subsurface zone⁶⁸. In contrast, the model simulation suggests that LDMPs with a size range below 10 μm can accumulate much heavier ballast (nearly 2.5 g cm^{-3}); however, the high vertical drag coefficient traps these LDMPs in the upper water column for a long time due to extremely slow settling velocity⁷¹. Although aggregation among LDMPs can accelerate the settling process, it is entirely by chance. It is positively correlated with the concentration of LDMPs in the water column, which further assists in forming the subsurface accumulation zone^{35,50,64}." (lines 248–259)

L160&L281: "It seems that..." this sentence is too conclusive, as other mechanisms are not tested, for example marine snow aggregation and fecal pellet sinking. These other mechanisms have been simply parameterized in another model and have similar results (mesopelagic accumulation), e.g. Kvale et al. (2020) <https://www.nature.com/articles/s41598-020-72898-4>. Likewise, biofouling was demonstrated to be able to distribute plastic particles

deep into the water column (hundreds of meters) in slow oscillations by Fischer et al. (2022) <https://doi.org/10.5194/bg-19-2211-2022>. Can you explain why your biofouling model does not behave similarly to that of Fischer et al.?

Response/action: The reviewer's concerns are well taken. In the revised manuscript, the statement has been improved.

"Microbially induced calcium carbonate precipitation may serve as one of the critical factors in depleting the LDMP concentrations at the ocean subsurface and enhancing the LDMP accumulation at the oceanic sediment¹⁶." (lines 119–121)

Moreover, our simulation without calcite precipitation does not perform like Fischer et al. because only biofouling is considered to make a fair comparison. Besides, biofouling cannot explain the broad detection of low-density microplastics in the ocean sediment, even with other oceanographic processes involved.

"The settling patterns of LDMPs in the deep ocean can be substantially altered by MICP, as compared to the results by Kooi's model, which only considered biofouling process (Fig. 1A)." (lines 106–108)

"Lobelle et al.³¹ and Fischer et al.²⁵ incorporated oceanographic processes with a settling model, including large-scale 3D advection, small-scale vertical turbulence, dynamic grazing, and wind-induced mixing, etc. The updated model allowed LDMPs to sink below the euphotic zone and mixed layer; however, only 15 of 10,000 LDMPs reached the ocean floor (>5,000 m) under favorable conditions²⁵." (lines 66–70)

Paragraph from L180. I really like this paragraph and associated Fig. 4. These are a highlight of the paper. Paragraph from L219. Are there observations that can be used to validate these simulated patterns? Perhaps Zhao et al. (2022) <https://doi.org/10.1111/gcb.16089>.

Response/action: The reviewer's concerns are well taken. In the revised manuscript, brief discussions regarding the simulated vertical sinking patterns of LDMPs are provided with the observation in the Arctic Ocean.

"The vertical distribution of LDMPs in the Arctic Ocean is unique compared to those in other oceans, and the present model can well explain this unique distribution of LDMPs. The concentration of LDMPs in the Arctic Ocean is higher at both the sea surface and oceanic sediment (seafloor) and lower in the upper water column (Fig. 5A)^{1,13,65}. No subsurface accumulation zone is observed, which could be attributed to the dramatic seasonal variations in light intensity and chlorophyll concentration⁷³. In summer, light intensity and chlorophyll concentration at the Arctic Ocean reach their maximums, and biofouling and MICP quickly uplift the density of LDMPs (Fig. 5B). Once overcoming the vertical drag coefficient, LDMPs can settle to the seafloor. The seawater density of the Arctic Ocean is almost consistent from the surface to the bottom (Fig. 5C), posing the least hindering effect on the settling of LDMPs⁷⁴.

There are also vertical downward currents in the Arctic Ocean⁷⁵, which further promote the settling of LDMPs, especially those with a size below 10 μm . In winter, however, light intensity is almost zero. LDMPs in the subsurface zone would ascend due to the lack of

biofouling and MICP, which explains the high abundances of LDMPs on the ocean surface¹ (Figs. 5D–5E). Seasonal alternation creates a relatively long period of time with no light, which means that the aggregation of LDMPs is insufficient due to inadequate ballast. As a result, the most favorable size range for vertical settling of LDMPs should be 10–500 μm based on the present model, which is consistent with the results from sediment core analyses^{12,13} (Fig 5F) that the most abundant LDMPs were in the 11–200 μm size class⁶⁵. The aging and fragmentation may also explain the absence of large LDMPs during long-range transport through the global ocean circulation^{61,67,76,77}." (lines 267–287)

Discussions: Kvale et al. (2020) <https://www.nature.com/articles/s41598-020-72898-4> Supplement had a whole sensitivity test (Figs. S6, S9) that demonstrated the role of biologically mediated (fecal pellet and marine snow aggregates) in sinking microplastic particles into the mesopelagic and in accumulation of plastic debris at these depths, but this is never discussed. Likewise, comparison to the Fischer et al. (2022) <https://doi.org/10.5194/bg-19-2211-2022> study and the Lobelle et al. (2021) <https://doi.org/10.1029/2020JC017098> study is not done.

Response/action: The reviewer's concerns are well taken. In the revised manuscript, additional discussions regarding the fecal pellet and marine snow aggregates are given. It should be noted that the calcite precipitation hypothesis is not an exclusive theory; instead it can combine with other oceanographic processes to reinforce the settling of LDMPs. We have added an extra discussion to describe these co-effects.

"However, the biofouling model encounters enormous challenges to explain the enrichment of LDMPs in the oceanic sediment. Modeled LDMPs were impossible to approach the ocean floor due to the continuing loss of biofilm²⁶. Fischer et al.²⁵ evaluated LDMP settling under another biofilm dynamic, assuming frustule attachment on LDMPs after settling breathe the twilight zone. Ingestion by marine organisms (fecal pellets) merely sinks a negligible fraction (0.13–0.19%) of LDMPs^{27,28}. Aggregation with biogenic particles (marine snow) could sink another 0.06–8.8% of LDMPs^{29,30}. Lobelle et al.³¹ and Fischer et al.²⁵ incorporated oceanographic processes with a settling model, including large-scale 3D advection, small-scale vertical turbulence, dynamic grazing, and wind-induced mixing, etc. The updated model allowed LDMPs to sink below the euphotic zone and mixed layer; however, only 15 of 10,000 LDMPs reached the ocean floor (>5,000 m) under favorable conditions²⁵." (lines 60–70)

"In previous studies, aggregation has been demonstrated as one of the important processes assisting the settling of ocean particles⁵⁰, which may also boost the settling of LDMPs with the size ranges of <10 μm and 1–5 mm^{35,64}. The simulation results indicate that such aggregation could sink a significant amount of LDMPs in the size range below 10 μm , aggregating them with portions of LDMPs in the size range of 1–5 mm. On one hand, oscillating large-sized LDMPs have more chances to collide with other LDMPs. On the other hand, small-sized LDMPs are much more abundant according to the exponential size-distribution of LDMPs. With high-density fouling, these small-sized LDMPs are the perfect ballast to aggregate with large-sized LDMPs. Aggregation of LDMPs with marine snow and

fecal pellets also occurs, and the aggregates could settle to the seafloor under a small vertical drag coefficient." (lines 220–229)

"The first accumulation zone occurs at the upper water column near the sea surface (60–400 m), coinciding with assumed static seawater conditions. Fischer et al.²⁵ and Lobelle et al.³¹ demonstrated that incorporating oceanographic processes such as large-scale 3D advection, small-scale vertical turbulence, dynamic grazing, and wind-induced mixing could spread LDMPs to much deeper waters (up to 5,500 m). Still, the maximum concentration of LDMPs remains in the upper water column⁶⁸. It seems that the oceanographic processes can assist in creating dynamic and complex scenarios for LDMPs." (lines 241–247)

Supersaturated dissolution of calcite is an observed phenomenon and responsible for the regulation of deep ocean calcite fluxes, e.g. Subhas et al. (2022) <https://doi.org/10.1029/2022GB007388>. Choosing to omit this process should be discussed.

Response/action: The reviewer's concerns are well taken. In the revised manuscript, the dissolution of calcite is added for consideration.

"The present model also considers the impact of calcite dissolution after LDMPs penetrate the calcite saturation depth (CSD)³². The degree of saturation of the seawater with respect to calcite is the driving force behind its precipitation or dissolution⁵⁴. The level of calcite saturation decreases in deep water. Because the solubility of calcite increases with increasing seawater depth/pressure, the concentration of dissolved inorganic carbon is elevated relative to total alkalinity levels in the deep ocean⁵⁵. Consequently, our model simulation indicates that all LDMPs suffer slight calcite loss and settling velocity deceleration after passing the CSD. However, the impact of calcite dissolution is relatively small for two reasons. First, the settling velocity of LDMPs is relatively high. Most LDMPs can reach the sea floor within less than one year, leading to insufficient dissolution. Second, the CSD varies regionally from 1,000 m to 4,600 m³². Calcite dissolution may never occur when the seafloor is above the CSD. For instance, the CSD in most regions of the Atlantic Ocean is below 3,500 m³². In comparison, the average depth of the Atlantic Ocean is 3,646 m." (lines 122–134)

Specific comments:

L61: From "Ingestion by seabirds..." This sentence can be deleted as it is not obviously relevant.

Response/action: The reviewer's concerns are well taken. In the revised manuscript, this sentence is removed as suggested.

L77: What is Fréchet similarity (please define)

Response/action: The reviewer's concerns are well taken. In the revised manuscript, Fréchet similarity has been removed since it is no longer be used.

L78: "The fitting performance..." this sentence is not justified by a comparison to a model without calcite ballast. Could model simulations that do not include calcite ballast be added to Fig. 1 panels?

Response/action: The reviewer's concerns are well taken. In the revised manuscript, the comparison between LDMPs with and without calcite precipitation is presented in Fig. 1.

L106: are ageing and fragmentation represented in this model? If not, please remove this sentence.

Response/action: The reviewer's concerns are well taken. In the revised manuscript, this sentence is removed as suggested.

L122: This sentence should be rephrased for clarity. I think 'resumption' is the wrong word. And is in situ CO₂ affecting calcification rate? No dissolution is included in this model-correct? (please state explicitly)

Response/action: The reviewer's concerns are well taken. In the revised manuscript, the sentence has been rephrased as reduced pH with references to support the statement.

"Although MICP also fades due to reduced pH caused by weakened algal photosynthesis^{32,53}, calcite precipitates are more difficult to detach from the plastic surface than biofilms." (lines 114–116)

A sentence added around L122 to describe how calcite precipitates are modelled to come off the plastic particle would be helpful for understanding this section.

Response/action: The reviewer's concerns are well taken. In the revised manuscript, the dissolution of calcite is added for consideration since quantitatively modeling the come off of calcite lacks data support.

"The present model also considers the impact of calcite dissolution after LDMPs penetrate the calcite saturation depth (CSD)³². The degree of saturation of the seawater with respect to calcite is the driving force behind its precipitation or dissolution⁵⁴. The level of calcite saturation decreases in deep water. Because the solubility of calcite increases with increasing seawater depth/pressure, the concentration of dissolved inorganic carbon is elevated relative to total alkalinity levels in the deep ocean⁵⁵. Consequently, our model simulation indicates that all LDMPs suffer slight calcite loss and settling velocity deceleration after passing the CSD. However, the impact of calcite dissolution is relatively small for two reasons. First, the settling velocity of LDMPs is relatively high. Most LDMPs can reach the sea floor within less than one year, leading to insufficient dissolution. Second, the CSD varies regionally from 1,000 m to 4,600 m³². Calcite dissolution may never occur when the seafloor is above the CSD. For instance, the CSD in most regions of the Atlantic

Ocean is below 3,500 m³². In comparison, the average depth of the Atlantic Ocean is 3,646 m." (lines 122–134).

Fig. 2: Adding biofouling only (no calcite) trajectories, velocities, and densities to these panels would enhance understanding of how calcite is uniquely affecting the results.

Response/action: The reviewer's concerns are well taken. In the revised manuscript, the comparison between LDMPs with and without calcite precipitation is presented in Fig. 1.

L152: "Due to low light intensity." the dominant control of light intensity on deep particle sinking is not clearly demonstrated by the results. Rephrasing to a density difference focus would be clearer.

Response/action: The reviewer's concerns are well taken. In the revised manuscript, the sentence is improved as suggested.

"Modeled LDMPs were impossible to approach the ocean floor due to the continuing loss of biofilm²⁶." (lines 61–62)

L176: "According to the simulation results..." this sentence does not make sense to me because the initial density cannot be modified by the settling time. Please revise.

Response/action: The reviewer's concerns are well taken. In the revised manuscript, the sentence is improved as suggested.

"Simulation results indicate that the settling of LDMPs exhibits essentially no relevance to the initial density of LDMPs (Figs. 3A–3C)." (lines 176–177)

Next sentence (L177) I am confused here because up to now I thought the plastic particles are not losing mass. Please explicitly state this somewhere before this point.

Response/action: The reviewer's concerns are well taken. In the revised manuscript, the sentence is improved as suggested.

"The accumulation of biofilm and MICP takes hours to days to complete, which is consistent with field observations⁵⁴. The density of settling LDMPs is mainly dependent on MICP and biofilm growth rather than the initial density." (lines 177–179)

Also L193, plastic is warped in the model by biofilm? I am confused.

Response/action: The reviewer's concerns are well taken. In the revised manuscript, the sentence is removed.

L214: "It seems that..." this sentence is self-contradictory. Could it be "maintain sinking" rather than "floating"? Or "small" should be "large"?

Response/action: The reviewer's concerns are well taken. The current statement is confusing and does not meet with the simulation. In the revised manuscript, the sentence is reorganized.

"In contrast, the vertical drag coefficient of small (< 500 μm) and non-spherical LDMPs (e.g., film, fragments, and fiber) is quite large, which could slow down the LDMP motion (Figs. 2C–2D). Established biofilms and calcite precipitates can be developed on the LDMP surface, leading to a much higher density of fouled LDMP than seawater density. In some cases, the density can grow up to 2.40 g cm^{-3} (Fig. 2D) before fouled LDMPs could leave the epipelagic zone. For fouled LDMPs with established biofilms and calcite precipitates, the vertical settling is straightforward, because the extra ballast of calcite could assist LDMPs to overcome seawater density increment and resist calcite dissolution (Fig. S1)." (lines 155–162)

L285: "If plastic debris..." Do you have a citation for this statement?

Response/action: The reviewer's concerns are well taken. In the revised manuscript, the statement is changed. We currently lack such an assessment, which means the statement comes from speculation.

"If LDMPs with calcite precipitates eventually sink to the seafloor, it might impact the ocean benthic ecosystem, global carbon cycle, and ocean calcite cycle." (lines 328–330)

Reviewer #3 (Remarks on code availability):

The code is insufficiently provided and with no documentation. I do not see how I would be able to reproduce the modelling from what is provided. As far as I can tell, only model outputs are provided.

Response/action: The reviewer's concerns are well taken. We will improve the code's readability and availability to make it fully functional.

References

- 1 Bergmann, M. & Klages, M. Increase of litter at the Arctic deep-sea observatory HAUSGARTEN. *Mar. Pollut. Bull.* **64**, 2734–2741, doi:<https://doi.org/10.1016/j.marpolbul.2012.09.018> (2012).
- 3 Yu, F., Yang, C., Zhu, Z., Bai, X. & Ma, J. Adsorption behavior of organic pollutants and metals on micro/nanoplastics in the aquatic environment. *Sci. Total Environ.* **694**, 133643, doi:<https://doi.org/10.1016/j.scitotenv.2019.133643> (2019).
- 5 Sorasan, C. *et al.* Ageing and fragmentation of marine microplastics. *Sci. Total Environ.* **827**, 154438, doi:<https://doi.org/10.1016/j.scitotenv.2022.154438> (2022).
- 12 Woodall, L. C. *et al.* The deep sea is a major sink for microplastic debris. *R. Soc. Open Sci.* **1**, 140317, doi:10.1098/rsos.140317 (2014).
- 13 Bergmann, M. *et al.* High quantities of microplastic in arctic deep-sea sediments from the HAUSGARTEN observatory. *Environ. Sci. Technol.* **51**, 11000–11010, doi:10.1021/acs.est.7b03331 (2017).
- 14 Peng, X. *et al.* Microplastics contaminate the deepest part of the world’s ocean. *Geochem. Perspect. Lett.* **9**, 1–5, doi:<https://doi.org/10.7185/geochemlet.1829> (2018).
- 15 Zhao, S. *et al.* Large quantities of small microplastics permeate the surface ocean to abyssal depths in the South Atlantic Gyre. *Glob. Change Biol.* **28**, 2991–3006, doi:<https://doi.org/10.1111/gcb.16089> (2022).
- 25 Fischer, R. *et al.* Modelling submerged biofouled microplastics and their vertical trajectories. *Biogeosciences* **19**, 2211–2234, doi:10.5194/bg-19-2211-2022 (2022).
- 26 Alldredge, A. L. & Gotschalk, C. In situ settling behavior of marine snow. *Limnol. Oceanogr.* **33**, 339–351, doi:<https://doi.org/10.4319/lo.1988.33.3.0339> (1988).
- 27 Desforges, J.-P. W., Galbraith, M. & Ross, P. S. Ingestion of microplastics by zooplankton in the Northeast Pacific Ocean. *Arch. Environ. Contam* **69**, 320–330, doi:10.1007/s00244-015-0172-5 (2015).
- 28 Davison, P. C. & Asch, R. G. Plastic ingestion by mesopelagic fishes in the North Pacific Subtropical Gyre. *Mar. Ecol. Prog. Ser.* **432**, 173–180 (2011).
- 29 Wang, X. *et al.* A review of microplastics aggregation in aquatic environment: Influence factors, analytical methods, and environmental implications. *J. Hazard. Mater.* **402**, 123496, doi:<https://doi.org/10.1016/j.jhazmat.2020.123496> (2021).
- 30 Kvale, K. F., Friederike Prowe, A. E. & Oschlies, A. A critical examination of the role of marine snow and zooplankton fecal pellets in removing ocean surface microplastic. *Front. Mar. Sci.* **6** (2020).
- 31 Lobelle, D. *et al.* Global modeled sinking characteristics of biofouled microplastic. *J. Geophys. Res. Oceans* **126**, e2020JC017098, doi:<https://doi.org/10.1029/2020JC017098> (2021).
- 32 Sulpis, O., Jeansson, E., Dinauer, A., Lauvset, S. K. & Middelburg, J. J. Calcium carbonate dissolution patterns in the ocean. *Nat. Geosci.* **14**, 423–428, doi:10.1038/s41561-021-00743-y (2021).

- 35 Long, M. *et al.* Interactions between microplastics and phytoplankton aggregates: Impact on their respective fates. *Mar. Chem.* **175**, 39–46, doi:<https://doi.org/10.1016/j.marchem.2015.04.003> (2015).
- 37 Molina Grima, E., Belarbi, E. H., Ación Fernández, F. G., Robles Medina, A. & Chisti, Y. Recovery of microalgal biomass and metabolites: process options and economics. *Biotechnology Advances* **20**, 491-515, doi:[https://doi.org/10.1016/S0734-9750\(02\)00050-2](https://doi.org/10.1016/S0734-9750(02)00050-2) (2003).
- 39 Heveran, C. M. *et al.* Engineered ureolytic microorganisms can tailor the morphology and nanomechanical properties of microbial-precipitated calcium carbonate. *Sci. Rep.* **9**, 14721, doi:10.1038/s41598-019-51133-9 (2019).
- 40 Dhama, N. K., Mukherjee, A. & Reddy, M. S. Micrographical, mineralogical and nano-mechanical characterisation of microbial carbonates from urease and carbonic anhydrase producing bacteria. *Ecol. Eng.* **94**, 443-454, doi:<https://doi.org/10.1016/j.ecoleng.2016.06.013> (2016).
- 41 Li, Y. *et al.* Preferential deposition of buoyant small microplastics in surface sediments of the Three Gorges Reservoir, China: Insights from biomineralization. *J. Hazard. Mater.* **468**, 133693, doi:<https://doi.org/10.1016/j.jhazmat.2024.133693> (2024).
- 42 Kooi, M. & Koelmans, A. A. Simplifying microplastic via continuous probability distributions for size, shape, and density. *Environ. Sci. Technol. Lett.* **6**, 551-557, doi:10.1021/acs.estlett.9b00379 (2019).
- 50 Rahmani, M., Gupta, A. & Jofre, L. Aggregation of microplastic and biogenic particles in upper-ocean turbulence. *Int. J. Multiph. Flow* **157**, 104253, doi:<https://doi.org/10.1016/j.ijmultiphaseflow.2022.104253> (2022).
- 51 Paul-Pont, I. *et al.* Constraints and priorities for conducting experimental exposures of marine organisms to microplastics. *Front. Mar. Sci.* **5**, doi:10.3389/fmars.2018.00252 (2018).
- 52 Tobias-Hunefeldt, S. *Community assembly drivers shift from bottom-up to top-down in a maturing in situ marine biofilm model* Master of Science thesis, University of Otago, (2020).
- 53 Plée, K., Pacton, M. & Ariztegui, D. Discriminating the role of photosynthetic and heterotrophic microbes triggering low-Mg calcite precipitation in freshwater biofilms (Lake Geneva, Switzerland). *Geomicrobiol. J.* **27**, 391-399, doi:10.1080/01490450903451526 (2010).
- 54 Mucci, A. The solubility of calcite and aragonite in seawater at various salinities, temperatures, and one atmosphere total pressure. *Am. J. Sci.* **283**, 780-799 (1983).
- 55 Millero, F. J. The thermodynamics of the carbonate system in seawater. *Geochim. Cosmochim. Ac.* **43**, 1651-1661, doi:[https://doi.org/10.1016/0016-7037\(79\)90184-4](https://doi.org/10.1016/0016-7037(79)90184-4) (1979).
- 56 Robert, O. The isoperimetric inequality. *Bull. Am. Math. Soc.* **84**, 1182-1238 (1978).
- 58 Weis, J. S. & De Falco, F. Microfibers: Environmental problems and textile solutions. *Microplastics* **1**, 626-639 (2022).
- 59 Takarina, N. D., Purwiyanto, A. I. S., Rasud, A. A., Arifin, A. A. & Suteja, Y. Microplastic abundance and distribution in surface water and sediment collected from

- the coastal area. *Glob. J. Environ. Sci. Manag.* **8**, 183-196, doi:10.22034/GJESM.2022.02.03 (2022).
- 61 Kvale, K., Prowe, A. E. F., Chien, C. T., Landolfi, A. & Oeschies, A. The global biological microplastic particle sink. *Sci. Rep.* **10**, 16670, doi:10.1038/s41598-020-72898-4 (2020).
- 62 Shanks, A. L. & Trent, J. D. Marine snow: sinking rates and potential role in vertical flux. *Deep-Sea Res. I: Oceanogr. Res. Pap.* **27**, 137-143, doi:[https://doi.org/10.1016/0198-0149\(80\)90092-8](https://doi.org/10.1016/0198-0149(80)90092-8) (1980).
- 63 Patonai, K., El-Shaffey, H. & Paffenhöfer, G.-A. Sinking velocities of fecal pellets of doliolids and calanoid copepods. *J. Plankton Res.* **33**, 1146-1150, doi:10.1093/plankt/fbr011 (2011).
- 64 Burd, A. B. & Jackson, G. A. Particle aggregation. *Annu. Rev. Mar. Sci.* **1**, 65–90, doi:10.1146/annurev.marine.010908.163904 (2009).
- 66 Galgani, L. *et al.* Hitchhiking into the deep: How microplastic particles are exported through the biological carbon pump in the north Atlantic Ocean. *Environ. Sci. Technol.* **56**, 15638-15649, doi:10.1021/acs.est.2c04712 (2022).
- 67 Bleck, R. An oceanic general circulation model framed in hybrid isopycnic-Cartesian coordinates. *Ocean Model.* **4**, 55-88, doi:[https://doi.org/10.1016/S1463-5003\(01\)00012-9](https://doi.org/10.1016/S1463-5003(01)00012-9) (2002).
- 68 Law Kara, L. *et al.* Plastic accumulation in the North Atlantic subtropical gyre. *Science* **329**, 1185–1188, doi:10.1126/science.1192321 (2010).
- 70 Cummings, J. A. Operational multivariate ocean data assimilation. *Quart. J. Royal Met. Soc., Part C* **131**, 3583-3604, doi:<https://doi.org/10.1256/qj.05.105> (2005).
- 71 Kiørboe, T., Tang, K., Grossart, H.-P. & Ploug, H. Dynamics of microbial communities on marine snow aggregates: colonization, growth, detachment, and grazing mortality of attached bacteria. *Appl. Environ. Microbiol.* **69**, 3036–3047, doi:10.1128/AEM.69.6.3036-3047.2003 (2003).
- 73 Berge, J. *et al.* In the dark: A review of ecosystem processes during the Arctic polar night. *Prog. Oceanogr.* **139**, 258-271, doi:<https://doi.org/10.1016/j.pocean.2015.08.005> (2015).
- 75 Timmermans, M.-L. & Marshall, J. Understanding Arctic Ocean circulation: A review of ocean dynamics in a changing climate. *Journal of Geophysical Research: Oceans* **125**, e2018JC014378, doi:<https://doi.org/10.1029/2018JC014378> (2020).
- 76 Onink, V., Kaandorp, M. L. A., van Sebille, E. & Laufkötter, C. Influence of particle size and fragmentation on large-scale microplastic transport in the Mediterranean Sea. *Environ. Sci. Technol.* **56**, 15528-15540, doi:10.1021/acs.est.2c03363 (2022).
- 81 Nabizadeh, R., Sajadi, M., Rastkari, N. & Yaghmaeian, K. Microplastic pollution on the Persian Gulf shoreline: A case study of Bandar Abbas city, Hormozgan Province, Iran. *Mar. Pollut. Bull.* **145**, 536-546, doi:<https://doi.org/10.1016/j.marpolbul.2019.06.048> (2019).
- 82 Andrady, A. L. Microplastics in the marine environment. *Mar. Pollut. Bull.* **62**, 1596–1605, doi:<https://doi.org/10.1016/j.marpolbul.2011.05.030> (2011).

REVIEWER COMMENTS

Reviewer #1 (Remarks to the Author):

The authors have comprehensively addressed the comments and made the required improvements. In particular, the revised manuscript provides a significantly more elaborated discussion by including necessary references from literature, which was one of my greatest concerns. In addition, the methods section, which was ultimately incomplete, is dramatically improved in the revised version. Finally, specific comments have also been addressed, however, I would advise the authors to closely review their manuscript for critical errors. For example, I doubt that the word "breathe" is intended in line 61.

Reviewer #1 (Remarks on code availability):

N/A

Reviewer #3 (Remarks to the Author):

The authors have greatly improved the quality of their manuscript since my last reading. There remain some grammar errors that I leave to the editors to correct if the manuscript is accepted.

Specific comments:

Line 25-26: Highlights the 'potential' importance...

Because without model validation with observations of calcite coated microplastics at depth, this study remains a sensitivity study

Line 61: 'breathe' should be 'beneath'

Line 63-64: Sentence on marine snow aggregation. Ref 30 show 0.06-8.8% sedimentary 'sequestration' potential, not sinking potential.

Kvale et al. 2020 <https://www.nature.com/articles/s41598-020-72898-4> shows that marine snow aggregation + fecal pellet removal can potentially remove all of the microplastics from the upper 50 m, but the model parameters are too poorly constrained to say anything more definitive.

Line 130: 'In our model' calcite dissolution may never occur...

I think the authors misunderstood my comment on supersaturated calcite dissolution, which is a major process in the upper ocean (see Subhas et al., <https://agupubs.onlinelibrary.wiley.com/doi/abs/10.1029/2022gb007388>). Calcite dissolution is better explained by a combination of oxygen consumption and saturation state than saturation state alone. This implies that calcite mediated LDMP transport is less effective than simulations based purely on thermodynamics would indicate.

Line 138: 'series of sensitivity analyses'

Line 148: 'Unlike the biofilm model' ... this is a key sentence, very important to highlight in the abstract. Please consider producing a schematic of this process, as well as those for smaller sized LDMPs.

Line 160: 'increment'-> 'the sw density gradient'

Line 174: 'Simulation results indicate that the initial density of LDMPs exhibits essentially no relevance to their settling, which is determined mainly by MICP and biofilm growth' (and then you can remove last sentence of paragraph)

Line 214: 'The high abundance...' Sentence is unclear. The high abundance of LDMPs in model sediments? Simulated water column?

Page 13: A schematic of the relevant processes in the Arctic could also be a very useful addition to a supplement, or to otherwise get onto the internet. I expect this would be a very useful resource for others.

Line 290: I think 'settled' is the wrong word. Maybe, 'reported'?

An off-track comment: you can potentially simulate a large storm, e.g. cyclone with your forcing data and see how this disrupts settling times. This could be very interesting looking at regions with consistently high pollution and regular seasonal storm tracks

Line 313-314: This is an important sentence with policy implications.

Fig 5A: What are the units? Please provide this information in the caption.

Reviewer #3 (Remarks on code availability):

There is no readme that I could find and I did not try to install and run the code. It was not clear how to do so.

Reviewer #1:

The authors have comprehensively addressed the comments and made the required improvements. In particular, the revised manuscript provides a significantly more elaborated discussion by including necessary references from literature, which was one of my greatest concerns. In addition, the methods section, which was ultimately incomplete, is dramatically improved in the revised version. Finally, specific comments have also been addressed, however, I would advise the authors to closely review their manuscript for critical errors. For example, I doubt that the word “breathe” is intended in line 61.

Response/action: Thank you very much for your positive feedback and suggestions. The reviewer’s concerns are well taken. The entire manuscript was subject to rigorous examination and check. Indeed, “breathe” was a typo. It should be “beneath” (line 62 in the revised manuscript).

Reviewer #1 (Remarks on code availability):

N/A

Reviewer #3:

The authors have greatly improved the quality of their manuscript since my last reading. There remain some grammar errors that I leave to the editors to correct if the manuscript is accepted.

Specific comments:

Line 25-26: Highlights the ‘potential’ importance...Because without model validation with observations of calcite coated microplastics at depth, this study remains a sensitivity study.

Response/action: The reviewer’s concerns are well taken. In the revised manuscript, the statement of “potential” has been highlighted as suggested.

“The present study employs a new model that considers **the potential** of an overlooked microbially induced calcium carbonate precipitation (MICP) process and new motion equations for irregular LDMPs.” (lines 17–19)

“Furthermore, LDMPs in the size range of 10–200 μm are most likely to gain sufficient density at the biofouling/MICP stage to independently sink to the ocean floor with relatively small drag coefficients, **potentially** explaining the selective enrichment of LDMPs in the oceanic sediment.” (lines 21–24)

Line 61: ‘breathe’ should be ‘beneath’

Response/action: Thank you for your careful checking our manuscript. Indeed, “breathe” was a typo. It should be “beneath” (line 62 in the revised manuscript).

Line 63-64: Sentence on marine snow aggregation. Ref 30 show 0.06-8.8% sedimentary ‘sequestration’ potential, not sinking potential.

Kvale et al. 2020 <https://www.nature.com/articles/s41598-020-72898-4> shows that marine snow aggregation + fecal pellet removal can potentially remove all of the microplastics from the upper 50 m, but the model parameters are too poorly constrained to say anything more definitive.

Response/action: The reviewer’s points are greatly appreciated. In the revised manuscript, the statement has been corrected as suggested.

“Aggregation with biogenic particles (marine snow) could **sequester** another 0.06–8.8% of LDMPs²².” (lines 64–65)

Line 130: ‘In our model’ calcite dissolution may never occur... I think the authors misunderstood my comment on supersaturated calcite dissolution, which is a major process in the upper ocean (see Subhas et al.,

<https://agupubs.onlinelibrary.wiley.com/doi/abs/10.1029/2022gb007388>). Calcite dissolution is better explained by a combination of oxygen consumption and saturation state than saturation state alone. This implies that calcite mediated LDMP transport is less effective than simulations based purely on thermodynamics would indicate.

Response/action: Thank you very much for your valuable suggestions. In the revised manuscript, the explanation of calcite dissolution has been revised as suggested. Moreover,

we merge the results of Subhas et al. within the revised manuscript with additional references.

“The present model also considers the impact of calcite dissolution during LDMPs settling, especially after penetrating the calcite saturation depth (CSD)²⁴. Seawater saturation state and oxygen consumption are the driving forces behind calcite precipitation or dissolution⁴⁰.” (lines 122–124)

“Oxygen consumption due to biological respiration generates carbon dioxide and organic acids, enhancing calcite dissolution⁴⁰.” (lines 127–129)

“First, oxygen consumption is largely inhibited during the MICP process because of the high pH environment near the LDMP surface created by photosynthesis²⁵. Second, the settling velocity of LDMPs is relatively high. Previous studies reported little CaCO₃ dissolution was observed on particles at a canonical sinking rate of 100 m day⁻¹, so the sinking flux at 4,000 m is almost identical to the surface flux⁴⁰. Based on our model simulation, most LDMPs can reach the sea floor within less than one year, leading to insufficient dissolution.” (lines 131–137)

Line 138: ‘series of sensitivity analyses’

Response/action: Thank you for catching the error. “analysis” has been revised to “analyse” (line 146 in the revised manuscript).

Line 148: ‘Unlike the biofilm model’... this is a key sentence, very important to highlight in the abstract. Please consider producing a schematic of this process, as well as those for smaller sized LDMPs.

Response/action: The reviewer’s points are well taken. In the revised manuscript, the abstract has been revised as suggested.

“The motion of LDMPs in the new model, exhibiting a damped oscillation pattern, is quite different from that in biofouling models.” (lines 19–21)

Line 160: ‘increment’-> ‘the sw density gradient’

Response/action: Thank you for your valuable suggestion. “increment” has been revised to “gradient” (line 169 in the revised manuscript).

Line 174: ‘Simulation results indicate that the initial density of LDMPs exhibits essentially no relevance to their settling, which is determined mainly by MICP and biofilm growth’ (and then you can remove last sentence of paragraph)

Response/action: The reviewer’s suggestions are greatly appreciated. In the revised manuscript, the statement has been revised as suggested and the last sentence of the paragraph has been deleted.

“Simulation results indicate that the settling of LDMPs exhibits essentially no relevance to the initial density of LDMPs, which is determined mainly by MICP and biofilm growth (Figs. 3A–3C).” (lines 183–185)

Line 214: ‘The high abundance...’ Sentence is unclear. The high abundance of LDMPs in model sediments? Simulated water column?

Response/action: The reviewer's concerns are well taken. In the revised manuscript, "in the simulated water column" has been inserted after "The high abundance of LDMPs" (line 222).

Page 13: A schematic of the relevant processes in the Arctic could also be a very useful addition to a supplement, or to otherwise get onto the internet. I expect this would be a very useful resource for others.

Response/action: The reviewer's points are well taken. We totally agree with the reviewer that adding a schematic of seasonal impacts in the Arctic would be useful for readers. We will integrate such schematics as an extend content and added as demo figures in the Github package at https://github.com/JePhyllis/MICP_Model_v1.0.

Line 290: I think 'settled' is the wrong word. Maybe, 'reported'?

Response/action: Thank you very much for catching the error. "settled" has been replaced with "reported" in the revised manuscript (line 298).

An off-track comment: you can potentially simulate a large storm, e.g. cyclone with your forcing data and see how this disrupts settling times. This could be very interesting looking at regions with consistently high pollution and regular seasonal storm tracks.

Response/action: Thank you very much for your valuable suggestions. We totally agree with the reviewer that adding extreme weather conditions not only assists exploring the variation of settling times, but also spurs further interests. We will integrate this good idea into future studies with existing models in the form of online websites or as an additional functional module.

Line 313-314: This is an important sentence with policy implications.

Response/action: Thank you for your confirmation of our perception. We will continue to emphasize policy implications in our future investigations.

Fig 5A: What are the units? Please provide this information in the caption.

Response/action: The reviewer's suggestions are well taken. In the revised manuscript, the units of Fig 5A have been added.

(A) Observed LDMPs Concentration (particles m⁻³) in the Seawater Column

(B) Global Chlorophyll Concentration (June 2022)

(C) Vertical Density Increment in the Global Oceans (June 2022)

(D) Global Chlorophyll Concentration (December 2021)

(E) Vertical Density Increment in the Global Oceans (December 2021)

(F) Sediment LDMPs	Abundance (particles L ⁻¹)	LDMPs Content (%)	Size (µm)	References
SW Pacific Ocean	200-2200	24	100-500	Peng et al. (2018)
Arctic Ocean	44-3463.71	8.2-20.8	<150 (99%)	Bergmann et al. (2017)
SW Indian Ocean	28-80	unknown	<100	Woodall et al. (2014)
NE Atlantic Ocean	120-800	unknown	<100	Woodall et al. (2014)
Mediterranean	200-700	unknown	<100	Woodall et al. (2014)
SW Pacific Ocean	40	unknown	75-161	Cauwenberghe et al. (2013)

Reviewer #3 (Remarks on code availability):

There is no readme that I could find and I did not try to install and run the code. It was not clear how to do so.

Response/action: The reviewer's points are well taken. The original code is uploaded to Github website (https://github.com/JePhyllis/MICP_Model_v1.0). The reviewers can download the entire model from the website and follow the instructions for model reproduction. The code package contains a readme file regarding step-by-step environment establishment, model simulation, and operation details.

REVIEWERS' COMMENTS

Reviewer #3 (Remarks to the Author):

The authors have satisfied my remaining concerns and I have no further comments on the manuscript.

Reviewer #3 (Remarks on code availability):

The authors have now provided clear instructions how to download and run their model code.